# SUPERVISED BINARY HYPERBOLIC EMBEDDINGS

## ABSTRACT

As datasets grow in size, vector-based search becomes increasingly challenging in terms of both storage and computational efficiency. Traditional solutions such as quantization techniques involve trade-offs between retrieval speed and accuracy, while hashing methods often require further optimization for binarization. In this work, we propose leveraging the compact nature of hyperbolic space for efficient search. Specifically, we introduce Binary Hyperbolic Embeddings, which transform complex hyperbolic similarity calculations into binary operations. We prove that these binary hyperbolic embeddings are retrieval-equivalent to their real-valued counterparts, ensuring minimal loss in retrieval quality. Our approach can be seamlessly integrated into FAISS to achieve improved memory efficiency and running speed while maintaining performance comparable to full-precision Euclidean embeddings. Notably, binary hyperbolic embeddings can also be combined with product quantization. We demonstrate significant improvements in storage efficiency, with a natural byproduct of speeding up, with scaling potential to larger datasets. A portion of the code is included in the supplementary materials, and the full implementation will be made publicly available.

## 1 INTRODUCTION

Compressed representations benefit information retrieval, as they greatly reduce index size, i.e., the memory requirements for data embeddings. Such compact property is desirable where retrieval-by-embedding needs to be fast or performed on large collections. Prior work has shown that considerable speed-ups can be obtained for Euclidean representations through binarization (Cai et al., 2020; Jacob et al., 2018; Kim et al., 2021), or by hashing Wang et al. (2018); Shen et al. (2020); Hoe et al. (2021b) representations on top of a network. These approaches do so by splitting the Euclidean representations into regular grids. In contrast, hyperbolic representations naturally allow for lower-dimensional representations (Long et al., 2020;

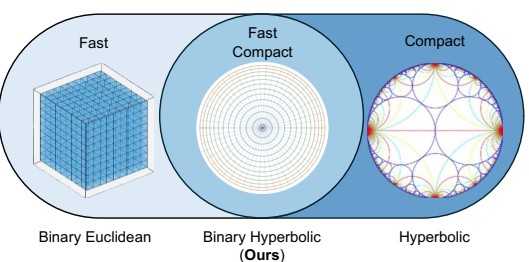

Figure 1: **Binary hyperbolic embeddings** aim for the best of both: the fast distance calculations of binary Euclidean embeddings and the compact representations of hyperbolic embeddings.

Ghadimi Atigh et al., 2021; Tseng et al., 2023) due to their compact nature. Such lower-dimensional representations allow retrieval systems for vastly reduced storage and have the potential to scale up. Unfortunately, hyperbolic compactness comes at the cost of hefty computations due to its complex metric (Peng et al., 2021). In this work, we overcome this complexity through binarization, unlocking the full potential of hyperbolic embeddings and pushing the embedding to even lower dimensionality. We show that it is possible to get the best of the fast distance calculations of binary embeddings and the compact representations of hyperbolic embeddings, as shown in Figure 1.

Hyperbolic deep learning is quickly gaining traction. Primarily, because it allows embedding hierarchies with minimal distortion (Nickel & Kiela, 2017), outperforming Euclidean hierarchical embeddings (Ganea et al., 2018b; Sala et al., 2018). These benefits have been shown for various research problems, from graph networks (Chami et al., 2019; Dai et al., 2021; Liu et al., 2019), reinforcement learning (Cetin et al., 2023) to large language models (Yang et al., 2024; Chen et al., 2024). Specifically, hyperbolic geometry allows for fewer embedding dimensionalities (Liu et al., 2020; Ermolov

et al., 2022; Tseng et al., 2023) and better hierarchical learning (Nickel & Kiela, 2017; Ganea et al., 2018b; Sonthalia & Gilbert, 2020). Despite these advantages, hyperbolic embeddings have not been a viable option for retrieval-by-embedding, as calculating the distance between embeddings involves complex vector operations [1].

This paper introduces supervised binary hyperbolic embedding, a binarization approach that addresses the core limitation of hyperbolic embeddings for retrieval. Our contributions are as follows:

- We unlock the speed potential of hyperbolic embeddings by proving that under retrieval, complex hyperbolic distance computation is ranking preserving to fast Hamming distance computation with our proposed binary encoding.
- Along with the inherent low dimensionality of hyperbolic space, we propose a natural binary hyperbolic embedding, which can obtain even lower-bit embedding with substantial speed-up with minimal loss in retrieval performance. Our embedding can be directly used with FAISS (Douze et al., 2024) for immediate memory reduction and speed-up.
- We show that these benefits hold across a variety of settings, including the ability to incorporate hierarchical knowledge and the potential to scale to larger retrieval sets.

Our work makes it possible to perform fast search in binarized hyperbolic space, making hyperbolic embeddings a viable supplement for large-scale search and retrieval.

## 2 RELATED WORK

Since search typically needs to occur on-the-fly (Yuan et al., 2020; Wang et al., 2018) or on huge collections (Jang & Cho, 2021; Chen et al., 2023), it is imperative to efficiently embed queries and data collections. The efficiency of an embedding can be expressed in bits, where fewer bits can ultimately only be obtained in two ways: using fewer embedding dimensions (Cao et al., 2020; Hausler et al., 2021) and/or using fewer bits per dimension (Choukroun et al., 2019; Yao et al., 2022; Bai et al., 2022).

**Hyperbolic low-dimensional representations** differ from the Euclidean representations for their ability to embed hierarchical structures with minimal distortion Ganea et al. (2018b); Tseng et al. (2023) due to the curved nature of the space Cannon et al. (1997). Most related, a variety of works find that hyperbolic space is naturally low-dimensional (Tifrea et al., 2019; Long et al., 2020; Shimizu et al., 2021; Ghadimi Atigh et al., 2021; Desai et al., 2023; Tseng et al., 2023). Such potential enables dimensionality reduction (Chami et al., 2021; Guo et al., 2022), but at the cost of computational overhead (Shimizu et al., 2021; Peng et al., 2021). In this paper, we exploit the low-dimensional nature, pushing it to fewer bits and turning the computational cost into acceleration.

**Low-bit embeddings.** For using fewer bits per embedding dimension, classical solutions are given by quantization techniques (Jégou et al., 2011; Jacob et al., 2018), benefiting quantization models designed for large-scale settings (Liu et al., 2021b; Yao et al., 2022). Inspired by these developments, we seek to bring the advantages of low-bit binarization to hyperbolic embeddings.

**Binary hashing embeddings** (Wang et al., 2017) learn compressed representations into compact binary codes (Shen et al., 2018b; Fan et al., 2020; Hoe et al., 2021b; Shen et al., 2020; Wei et al., 2024) while preserving the semantic similarity (Yuan et al., 2020) or structure (Li & van Gemert, 2021) of the original data, thereby reducing storage and computational costs. Unlike our proposed binarization approach, hashing requires NP-hard optimization and thus demands either complex optimization (Shen et al., 2018a) or substantial relaxation (Wei et al., 2024).

**Hyperbolic isometry**. There are five isometric models for hyperbolic space (Cannon et al., 1997). We focus on the Poincaré disk model as it is highly suited for binarization; the coordinates on the Poincaré ball are finite, and each axis is symmetric, allowing us to use a simple binarization strategy. Further discussion on the other hyperbolic models can be found in Appendix A.2.

**Approximate Nearest Neighbor** (ANN) search significantly reduces query time for large-scale retrieval (Muja & Lowe, 2014; Douze et al., 2024), where the exact nearest neighbor search is com-

---

[1]We quantify the complexity of hyperbolic distance $d_{\mathbb{D}}(\cdot, \cdot)$ in Appendix D.

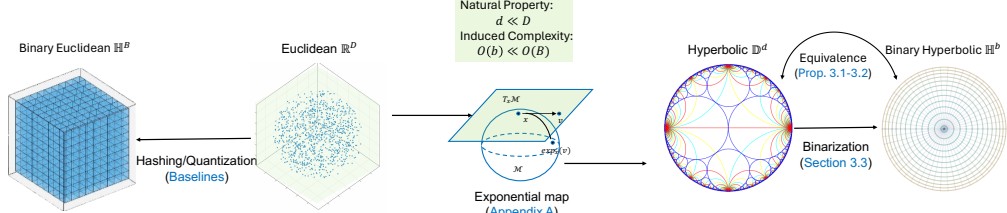

Figure 2: **Overview of our approach.** Where existing baselines perform quantization in Euclidean space **(left)**, we construct binary representations possible in hyperbolic space **(right)**. The main contribution (Propositions 3.1-3.2) highlights the equivalence between hyperbolic embedding $\mathbb{D}^d$ and its corresponding binary hyperbolic embedding $\mathbb{H}^b$. Section 3.3 details the binarization process.

putationally prohibitive. Classical approaches to ANN include tree-based methods (Arya & Mount, 1998), graph-based methods (Malkov & Yashunin, 2018), product quantization based (Jégou et al., 2011; Ge et al., 2014) to balance accuracy and speed. More application-oriented, DiskANN (Subramanya et al., 2019) combines in-memory computation with a disk-resident graph. enabling billion-scale nearest neighbor search on a single machine. In contrast, our approach efficiently performs **exact nearest neighbour search**, by leveraging compact binary hyperbolic embeddings, we avoid the compromises inherent in ANN thus eliminating reliance on distance gap preservation.

## 3 BINARY HYPERBOLIC EMBEDDING

In this work, we strive to find a binarizer $g(\cdot)$ which encodes real-valued hyperbolic embeddings $\boldsymbol{x}$ into a binary format $\boldsymbol{x}^b$, such that $\boldsymbol{x}^b = g(\boldsymbol{x})$. We do so by constructing an approximate equivalence between hyperbolic distance $\mathrm{d}_{\mathbb{D}}(\cdot, \cdot)$ based search and Hamming distance $\mathrm{d}_{\mathbb{H}}(\cdot, \cdot)$ based search. Then, using the Poincaré model for simplicity, we show how to binarize distances in hyperbolic space[2].

### 3.1 PROBLEM STATEMENT

Given a query embedding $\boldsymbol{q}$, retrieval is performed through a nearest neighbor search in the database $\mathcal{D}$ that is represented on manifold $\mathcal{M}$:

$$\underset{\boldsymbol{x} \in \mathcal{D}}{\arg \min} \, \mathrm{d}_{\mathcal{M}}(\boldsymbol{q}, \boldsymbol{x}). \tag{1}$$

This work argues for hyperbolic geometry as the manifold of choice for retrieval. Its effectiveness, especially in low-dimensional settings, is however offset by the computation complexity of hyperbolic distance $\mathrm{d}_{\mathbb{D}}(\boldsymbol{q}, \boldsymbol{x})$ curvature $c$:

$$\mathrm{d}_{\mathbb{D}}(\boldsymbol{q}, \boldsymbol{x}) = \frac{2}{\sqrt{c}} \tanh^{-1} \left( \sqrt{c} \left\| \frac{\left(1 - 2c\langle \boldsymbol{q}, \boldsymbol{x} \rangle + c\|\boldsymbol{x}\|^2\right) \boldsymbol{q} + \left(1 - c\|\boldsymbol{q}\|^2\right) \boldsymbol{x}}{1 - 2c\langle \boldsymbol{q}, \boldsymbol{x} \rangle + c^2\|\boldsymbol{q}\|^2\|\boldsymbol{x}\|^2} \right\|^2 \right). \tag{2}$$

In this work, we tackle the complexity of hyperbolic metric through binarization of the hyperbolic space as $\boldsymbol{x}^b = g(\boldsymbol{x})$, where we largely benefit from low-dimensional representation while speeding up distance calculations.

**Ranking Preservation.** To demonstrate the potential of binary hyperbolic embeddings for retrieval, we prove the ranking preservation property between binary hyperbolic embeddings and real-valued hyperbolic embeddings. In other words:

$$\mathrm{d}_{\mathbb{H}}(\boldsymbol{q}^b, \boldsymbol{x}_1^b) \leq \mathrm{d}_{\mathbb{H}}(\boldsymbol{q}^b, \boldsymbol{x}_2^b) \Rightarrow \mathrm{d}_{\mathbb{D}}(\boldsymbol{q}, \boldsymbol{x}_1) \leq \mathrm{d}_{\mathbb{D}}(\boldsymbol{q}, \boldsymbol{x}_2). \tag{3}$$

In this paper, we connect the Hamming metric to the hyperbolic metric via a (conditioned) ranking preservation between the hyperbolic metric and Euclidean metric on the hyperbolic embeddings.

---

[2]We provide preliminaries of related hyperbolic properties in Appendix A

## 3.2 FROM HYPERBOLIC TO HAMMING

We link the hyperbolic distance to the Hamming distance in three steps: (i) we first prove the ranking preservation between hyperbolic and Euclidean metric up under certain conditions, and (ii) using Euclidean metric as a bridge, we show the equivalence between Hamming distance (over binary hyperbolic embeddings) and the hyperbolic distance (over real-valued hyperbolic embeddings).

Proposition 3.1 states that for hyperbolic embeddings, under retrieval conditions, it is possible to produce the same retrieval output between hyperbolic metric $d_{\mathbb{D}}(\cdot, \cdot)$ and Euclidean metric $d_{\mathbb{R}}(\cdot, \cdot)$.

**Proposition 3.1.** *(Ranking Preservation) Given query $q \in \mathbb{D}^d$, For any $x_1, x_2 \in \mathbb{D}^d$ in the database such that $|\|x_1\| - \|x_2\|| = \epsilon$, given the Poincaré disk boundary tolerance margin $\delta$, if $0 \leq \epsilon \leq \delta \frac{d_{\mathbb{R}}(q, x_2) - d_{\mathbb{R}}(q, x_1)}{d_{\mathbb{R}}(q, x_2)}$, then $d_{\mathbb{R}}(q, x_1) \leq d_{\mathbb{R}}(q, x_2)$ implies $d_{\mathbb{D}}(q, x_1) \leq d_{\mathbb{D}}(q, x_2)$.*

*Proof.* The proof is in Appendix B.2. $\square$

This proposition demonstrates that under the condition $0 \leq \epsilon \leq \delta \frac{d_{\mathbb{R}}(q, x_2) - d_{\mathbb{R}}(q, x_1)}{d_{\mathbb{R}}(q, x_2)}$, the hyperbolic distance produces the same retrieval output as Euclidean distance. It is important to note that the precondition $\|\|x_1 - x_2\| = \epsilon \leq \delta \frac{d_{\mathbb{R}}(q, x_2) - d_{\mathbb{R}}(q, x_1)}{d_{\mathbb{R}}(q, x_2)}$ holds not only because normalizing embeddings is a common practice (Radford et al., 2021), but also because the embeddings used in this paper are closely distributed around normalized prototypes (Long et al., 2020; Kasarla et al., 2022).

As a follow-up step, in Proposition 3.2 we prove that under proper binarization $g(x) = x^b$, hyperbolic metric $d_{\mathbb{D}}(x, y)$ yields identical retrieval outputs as Hamming metric $d_{\mathbb{H}}(x^b, y^b)$:

**Proposition 3.2.** *(Binary Ranking Preservation) For a binarizer $g(\cdot)$ such that $\langle x^+, y^+ \rangle \propto \langle g(x), g(y) \rangle$, $d_{\mathbb{D}}(x, y)$ yields the same ranking results as to Hamming distance $d_{\mathbb{H}}(x^b, y^b) = \|x^b \oplus y^b\|_0$ for nearest neighbor search.*

*Proof.* The proof is in Appendix B.3. $\square$

Intuitively, the propositions state that a hyperbolic distance-based search generates the same retrieval ordering as a Hamming distance-based search, which can be computed quickly through binary operations. This is exactly the step to make distance-based nearest neighbor search fast in hyperbolic embeddings. Throughout this work, we use the Poincaré ball for our hyperbolic operations due to its widespread use in deep learning (Peng et al., 2021; Mettes et al., 2023), but we note that our approach applies to any hyperbolic model:

*Remark* 3.3. Under the isometry defined in Cannon et al. (1997), five hyperbolic models yield equivalent retrieval results.

*Proof.* The proof is in Appendix B.4. $\square$

## 3.3 BINARY HYPERBOLIC QUANTIZATION

Based on Proposition 3.2, we can perform binary operation-based search while using hyperbolic embeddings. To make this practically operational, we first generate full-precision hyperbolic embeddings on the Poincaré ball by optimizing embedding network $f(\cdot)$ [3] and then binarize the embeddings via quantization.

**Binary quantization**. With the trained hyperbolic embedding $x$, we perform binary quantization to obtain the binary representation of $x$, denoted as $x^b = g(x)$. In this section, we show that by designing a binarizer $g(\cdot)$ such that it satisfies $\langle g(x), g(y) \rangle \propto \langle x, y \rangle$ in a block-wise manner, we can exploit Proposition 3.2 for a binary Hamming distance-based search with hyperbolic embeddings.

In the Poincaré ball model, all dimensions fall in the radius of the ball $(-r, r)$. We shift each dimension by $r$ to make it in the range $(0, 2r)$:

$$x^+ = x + r. \tag{4}$$

---
[3]Hyperbolic embedding generation is described in Appendix B

This shift simplifies the calculations without changing the Euclidean distance:

$$\mathrm{d}_{\mathbb{R}}(\boldsymbol{x}^+, \boldsymbol{y}^+) = \|\boldsymbol{x}^+ - \boldsymbol{y}^+\| = \|\boldsymbol{x} + r\mathbf{1} - (\boldsymbol{y} + r\mathbf{1})\|$$
$$= \|\boldsymbol{x} - \boldsymbol{y}\| = \mathrm{d}_{\mathbb{R}}(\boldsymbol{x}, \boldsymbol{y}). \tag{5}$$

In our proposed approach, the representation undergoes a dimension-wise quantization process. For $n$ bits used by each dimension, we partition each dimension into a distinct set of $2^n - 1$ quantization levels under the same framework as Jeon et al. (2020) with respect to scale:

$$s = \frac{\sup(\boldsymbol{x}) - \inf(\boldsymbol{x})}{2^n - 1} = \frac{\sup(\boldsymbol{x}^+) - 0}{2^n - 1} = \frac{2r}{2^n - 1}, \tag{6}$$

where $\sup(\cdot)$ is the supremum and $\inf(\cdot)$ is the infimum. Then we can convert each dimension concerning the scale into integers, which can be converted into *n-bits* binary code:

$$\boldsymbol{x}_{\mathrm{int}} = \lfloor \frac{\boldsymbol{x}^+}{s} \rceil = \sum_{i=1}^{n} 2^{n-i} \cdot \boldsymbol{x}_i^b = 2^{n-1} \cdot \boldsymbol{x}_1^b + 2^{n-2} \boldsymbol{x}_2^b + \cdots + 2^0 \boldsymbol{x}_n^b, \tag{7}$$

where $\boldsymbol{x}_i^b \in \{0, 1\}^d$ represent the binary code for $i$-th significant bits in each dimension of $\boldsymbol{x}_{\mathrm{int}}$.

To relate $\boldsymbol{x}^b$ to $\boldsymbol{x}^+$, we can similarly decompose vector $\boldsymbol{x}$ as a summation of base vectors that correspond to different significant bits[4]:

$$\boldsymbol{x}^+ = \boldsymbol{x}_1^+ + \boldsymbol{x}_2^+ + \cdots + \boldsymbol{x}_n^+.$$

**Binary grouping**: assuming zero quantization error, we have a **block-wise proportionals**:

$$\langle \mathbf{x}^+, \mathbf{y}^+ \rangle \propto \langle \mathbf{x}_{int}^+, \mathbf{y}_{int}^+ \rangle$$
$$\langle \mathbf{x}_1^+, \mathbf{y}_1^+ \rangle \propto \langle \mathbf{x}_1^b, \mathbf{y}_1^b \rangle$$
$$\langle \mathbf{x}_2^+, \mathbf{y}_2^+ \rangle \propto \langle \mathbf{x}_2^b, \mathbf{y}_2^b \rangle$$
$$\cdots$$
$$\langle \mathbf{x}_n^+, \mathbf{y}_n^+ \rangle \propto \langle \mathbf{x}_n^b, \mathbf{y}_n^b \rangle, \tag{8}$$

which results in:

$$\langle \boldsymbol{x}^+, \boldsymbol{y}^+ \rangle = s^2 \langle \boldsymbol{x}_{\mathrm{int}}, \boldsymbol{y}_{\mathrm{int}} \rangle = 4^{n-1} \cdot \langle \boldsymbol{x}_1^b, \boldsymbol{y}_1^b \rangle + 4^{n-2} \cdot \langle \boldsymbol{x}_2^b, \boldsymbol{y}_2^b \rangle + \cdots + \langle \boldsymbol{x}_n^b, \boldsymbol{y}_n^b \rangle, \tag{9}$$

leading to a **block-wise application** of Proposition 3.2. We can obtain the distance metric for binary hyperbolic embeddings as:

$$\mathrm{d}_{\mathbb{R}}(\boldsymbol{x}, \boldsymbol{y}) = \mathrm{d}_{\mathbb{R}}(\boldsymbol{x}^+, \boldsymbol{y}^+) \propto d_{\mathbb{H}}^{\Sigma}(\boldsymbol{x}^b, \boldsymbol{y}^b) = \Sigma_{i=1}^{n} 4^{n-i} \cdot \mathrm{d}_{\mathbb{H}}(\boldsymbol{x}_i^b, \boldsymbol{y}_i^b) \tag{10}$$

where $d_{\mathbb{H}}^{\Sigma}(\boldsymbol{x}^b, \boldsymbol{y}^b)$ is a summation of the scaled hamming distance, hence we can use the scaled binary hamming distance as an approximation of the real-valued distance. The scaling only happens on each of the $n-1$ bits, resulting in $n-1$ binary bit-shift operations with integer addition, which can be efficiently carried out. Equipped with a hyperbolic embedding network $f(\cdot)$ and binarization $g(\cdot)$, fast retrieval can be performed by embedding the entire database. Then for a query $q$ and search collection $S$, both embedded to $\mathbb{D}$ and quantized, we can perform fast nearest neighbor search:

$$\arg\min_{\boldsymbol{v} \in S} d_{\mathbb{H}}^{\Sigma}(\boldsymbol{q}^b, \boldsymbol{v}^b) \stackrel{\triangle}{=} \arg\min_{\boldsymbol{v} \in S} \mathrm{d}_{\mathbb{D}}(\boldsymbol{q}, \boldsymbol{v}). \tag{11}$$

## 4 EXPERIMENTS

### 4.1 EXPERIMENTAL SETUP

We focus on retrieval in the image and video domains to measure the performance of various embedding compression approaches across a range of compression levels for retrieval performance and speed. hyperbolic LLMs are not included as they are either not yet open-sourced (Chen et al., 2024) or the text embeddings stay in Euclidean space (Yang et al., 2024). The performance is measured

---

[4]We show a concrete example in Appendix

Table 1: **Comparing manifolds and binarization for retrieval** on CIFAR100, ImageNet1K, and Moments-in-Time. Underlined scores denote best full-precision embedding performance; **bold** scores denote best binary embedding performance. For each manifold, we use the following settings: Euclidean space $\mathbb{R}^D$ (Liu et al., 2021a), hyperspherical space $\mathbb{S}^d$ (Kasarla et al., 2022), and hyperbolic space $\mathbb{D}^d$ (Long et al., 2020). $D, d$ denotes the dimensionality of the real-valued space, and $B, b$ is the binary representation's dimensionality. All retrievals are cut off @50. With full precision, hyperbolic embeddings already outperform Euclidean embeddings but are slow to evaluate. Our binary hyperbolic embeddings at 512 bits can maintain this performance while being much faster to evaluate, thereby maintaining performance at a much smaller embedding size.

| Manifold | Embedding size | CIFAR100 mAP | SmAP | Speed | ImageNet1K mAP | SmAP | Speed | Moments-in-Time mAP | SmAP | Speed |
|---|---|---|---|---|---|---|---|---|---|---|
| $\mathbb{R}^D$ | 16384 bits | 0.7938 ±0.0014 | 0.8868 ±0.0029 | 1.00x | 0.6324 ±0.0037 | 0.6879 ±0.0035 | 1.00x | 0.1598 ±0.0002 | 0.2371 ±0.0025 | 1.00x |
| $\mathbb{S}^d$ | 8192 bits | 0.8110 ±0.0037 | 0.8953 ±0.0021 | 0.99x | 0.6314 ±0.0019 | 0.6870 ±0.0021 | 1.01x | 0.1624 ±0.0015 | 0.2408 ±0.0022 | 1.00x |
| $\mathbb{D}^d$ | 8192 bits | 0.8078 ±0.0015 | 0.9017 ±0.0014 | 0.22x | 0.6344 ±0.0011 | 0.6894 ±0.0010 | 0.18x | 0.1780 ±0.0001 | 0.2578 ±0.0019 | 0.21x |
| $\mathbb{R}^B$ | 1024 bits | 0.6127 ±0.0005 | 0.7827 ±0.0002 | 2.21x | 0.6299 ±0.0008 | 0.6840 ±0.0076 | 2.28x | 0.1387 ±0.0013 | 0.2080 ±0.0017 | 2.29x |
| $\mathbb{S}^b$ | 512 bits | 0.7943 ±0.0006 | 0.8847 ±0.0004 | 4.19x | 0.6320 ±0.0015 | 0.6863 ±0.0017 | **8.25x** | 0.1584 ±0.0018 | 0.2303 ±0.0019 | 4.47x |
| $\mathbb{D}^b$ (ours) | 512 bits | **0.7948** ±0.0023 | **0.9014** ±0.0007 | **4.20x** | **0.6358** ±0.0018 | **0.6992** ±0.0017 | 8.24x | **0.1769** ±0.0015 | **0.2536** ±0.0018 | **4.50x** |

Table 2: **Integrating Binary Hyperbolic Embeddings in FAISS** on ImageNet1K-val. The results show that we can directly incorporate our approach into FAISS, making for a retrieval method that is strong in performance, with minimal index size, and top retrieval speed.

| Embedding | mAP@50 ↑ | SmAP@50 ↑ | Index Size ↓ | Retrieval Time (s) ↓ |
|---|---|---|---|---|
| Euclidean-512D | 0.6324 ±0.0037 | 0.6879 ±0.0035 | 102MB | 2.83 ± 0.07 |
| Euclidean-256D | 0.5847 ±0.0004 | 0.6374 ±0.0003 | 51MB | 1.11 ± 0.04 |
| Hyperbolic-256D | 0.6344 ±0.0011 | 0.6894 ±0.0010 | 51MB | 1.11 ± 0.06 |
| BinaryHyperbolic-256bit | 0.6320 ±0.0014 | 0.6875 ±0.0015 | **1.5MB** | **0.28 ± 0.03** |
| BinaryEuclidean-512bit | 0.5849 ±0.0024 | 0.6376 ±0.0015 | 3MB | 1.03 ± 0.04 |
| BinaryHyperbolic-512bit | **0.6358** ±0.0006 | **0.6992** ±0.0005 | 3MB | 1.04 ± 0.03 |

with mean average precision (mAP@50) and speed as the relative difference in retrieval time in seconds on the test set.

**Datasets.** We use three well-studied datasets with optional hierarchical knowledge: CIFAR100, that comes with an officially defined hierarchy (Krizhevsky et al., 2009), while for ImageNet1K each class is a node in the WordNet hierarchy (Miller, 1995).In Moments in Time, each class is a node in the VerbNet hierarchy (Schuler, 2006). We also examine the large-scale Quick Draw dataset (Google, 2023), which contains 50 million sketch images from 345 categories. There is no hierarchy in Quick Draw, we simply regard all categories as belonging to a super-class *root*.

**Implementation details.** For hyperbolic prototpes learning, we use a curvature of $c = 0.1$ and the Riemannian Adam optimizer (Becigneul & Ganea, 2019), supported by the *geoopt* (Kochurov et al., 2020) with a learning rate $10^{-4}$. In practice, when learning hyperbolic embeddings based on hyperbolic prototypes, Riemannian Adam can be replaced by Adam as the learnable parameters are in Euclidean space. Unless specified otherwise, we report supervised results on hyperbolic embeddings with hierarchical prototypes. All experiments were performed on a single Nvidia A6000 GPU. For the image experiments, we use the Swin (Liu et al., 2021a) on $32 \times 32$ patches for CIFAR-100 and ImageNet1K. For the video experiments, we use a pre-trained 3D Swin (Liu et al., 2022).

**Evaluation.** For a fair comparison, the same frozen backbone is used across all competing models. Unless stated otherwise, we use two bits per dimension for binarization following Hubara et al. (2018). We use commonly used mAP metric for evaluating retrieval, we also use SmAP similar to Long et al. (2020); Ghadimi Atigh et al. (2021) which takes into account the proximity in the class hierarchy for retrieved items. Specifically, when an item retrieved is just one hop away (i.e., same parent class) from the ground truth it is considered a true positive. To measure the speed-up, we perform both FAISS-based evaluation and purpose-made stand-alone experiments for measuring the retrieval speed. Note that Speed-up relies on the implementation in the mathematical library used. For example, a boolean variable in Pytorch is treated as an 8-bit unsigned int, which does not accurately reflect speed-up. Therefore, in addition to FAISS, we use a C++ implementation that supports vectorized float&bitwise operations to evaluate the speed-up. All speed-ups are relative to 512-dimensional full-precision Euclidean representation.

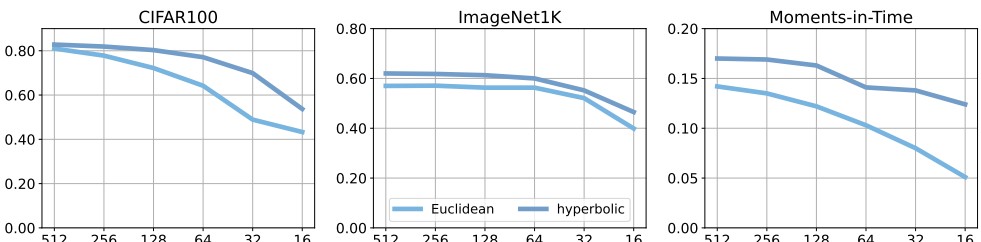

Figure 3: **Retrieval performance (mAP@50) as a function of bits** on CIFAR100, ImageNet1K, and Moments-in-Time. Binary hyperbolic embeddings allow for strong compression while maintaining performance, especially in low-bit settings.

## 4.2 BINARY VERSUS NON-BINARY RETRIEVAL

We first investigate the effect of manifold and binarization in Table 1 on retrieval performance and speed. We use 512 bits for binarized embeddings and report the results for mAP@50. As Equation 11 shows that our binarization-based similarity is equivalent to the similarity in $\mathbb{R}$ and $\mathbb{D}$, we can use the same binarization strategy across all three manifolds: Euclidean $\mathbb{R}$, hyperspherical $\mathbb{S}$, and hyperbolic $\mathbb{D}$. We use the same frozen backbone with a linear projection on top, supervised by the retrieval task, to get features with the same dimensionality. The Euclidean baseline follows conventional cross-entropy optimization, while the hyperspherical baseline uses maximally separate prototypes (Kasarla et al., 2022) optimized with cosine similarity between all prototype pairs. The hyperbolic embedding is trained akin to Long et al. (2020).

The results in Table 1 show that for full-precision embeddings, hyperbolic space shows great promise for retrieval, outperforming its Euclidean and hyperspherical alternatives. However, hyperbolic embedding retrieval is five times slower than Euclidean retrieval. With binary hyperbolic embeddings, we can induce a 2.2× to 8.2 × speed-up with the highest retrieval scores. Hyperbolic embeddings retain good performance when binarized, highlighting the strong match between hyperbolic space and binarization. Results in Table 2 also show its effectiveness when combined with FAISS, where our binary hyperbolic embedding can be seamlessly integrated into this common libary.

## 4.3 EFFECT OF BIT LENGTH AND QUANTIZATION LEVEL

**Effect of bit length.** To explore the impact of low-bits embeddings for fast retrieval, we compare Euclidean to hyperbolic embeddings as a function of the number of bits, as shown in Figure 3. For both geometry, we investigate using 512, 256, 128, and 64 bits, corresponding to 256-, 128-, 64-, and 32-dimensional embedding dimensions.

Figure 4 shows the trade-off between embedding dimensions and quantization levels on ImageNet1K. Larger, darker bubbles represent configurations with more bits and better mAP but slower speed. The baseline is 512d × 32 bits with an mAP of 0.6344 and no speedup (1×). Smaller, lighter bubbles show faster configurations, like 128d × 2 bits, achieving 16× speedup with an mAP of 0.6365. The figure illustrates that binary hyperbolic embeddings offer significant speedup (up to 8×) while maintaining comparable performance to the full-precision baseline.

Table 3: **Trade-off Between embedding dimensions and quantization bits** on ImageNet1K. Underlined scores denote full precision. Binary hyperbolic embeddings accelerate largely with roughly the same performance. The speed-up is based on integrating our embedding in Faiss. Another C++ version speed-up analysis is in Figure 4

| $dim \times bits$ | mAP@50 ↑ | Speed(s) ↓ |
|---|---|---|
| 8 ×2 | 0.1641 ±0.0011 | 0.46 ±0.03 |
| 8 ×4 | 0.1384 ±0.0005 | 0.22 ±0.08 |
| 16 ×2 | 0.2832 ±0.0008 | **0.17** ±0.02 |
| 16 ×4 | 0.2745 ±0.0009 | 0.18 ±0.02 |
| 32 ×2 | 0.4817 ±0.0072 | 0.18 ±0.02 |
| 32 ×4 | 0.4479 ±0.0079 | 0.22 ±0.02 |
| 64 ×2 | 0.5814 ±0.0104 | 0.21 ±0.02 |
| 64 ×4 | 0.5968 ±0.0022 | 0.21 ±0.01 |
| 128×2 | 0.6365 ±0.0067 | 0.28 ±0.02 |
| 128×4 | 0.6320 ±0.0014 | 1.03 ±0.04 |
| 256×2 | **0.6358** ±0.0006 | 1.00 ±0.02 |
| 512×32 | 0.6344±0.0036 | 2.83 ±0.07 |

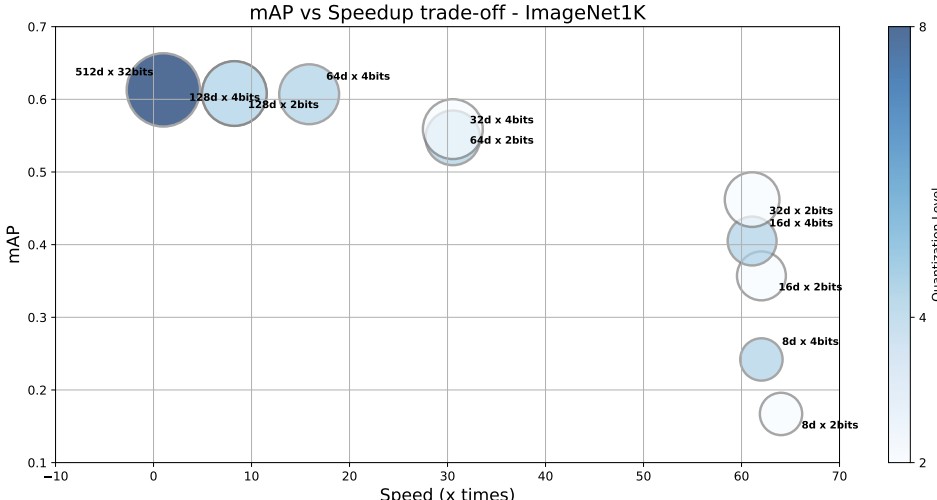

Figure 4: **The effect of embedding dimensions and quantization bits** on ImageNet1K-Val. The darkest bubble denotes the full-precision Euclidean embedding. With binary hyperbolic embeddings, we can obtain significant ($\sim 8\times$) acceleration at roughly the same performance.

**Effect of quantization level.** By quantizing each dimension into levels we can choose the bits per dimension. The total number of bits can therefore be determined by using more embedding dimensions with few bits or vice versa. For example, a 128-bits = 64d $\times$ 2 bits, or 32d $\times$ 4 bits. In Table 3, we show the impact across multiple choices of bit sizes. Overall, more embedding dimensions with stronger compression perform better than the other way around. We show more comparisons of other datasets in Appendix D.1.

The speed-up results in Table 3 paint a clear picture: the speed-up can be obtained without hampering retrieval performance. Our approach allows for much bigger speed-up, but high compression then comes at the price of lower retrieval performance, making it a design choice how to balance both.

### 4.4 EFFECT OF HIERARCHICAL KNOWLEDGE

A key benefit of hyperbolic space is the capability to embed hierarchical knowledge with minimal distortion for hierarchical embeddings (Ganea et al., 2018b; Sala et al., 2018). Such property enables a hyperbolic network to retrieve semantically similar items of adjacent classes (Long et al., 2020), Following which setting, in Appendix D.2 we report results with hierarchical knowledge-aware retrieval. We can maintain retrieval performance in both standard and hierarchy-aware settings. Even with the lowest number of bit lengths.

**Qualitative analysis.** In Figure 5 we compare a non-hierarchical spherical space with our hierarchical hyperbolic space. All classes are connected to classes with pairwise cosine similarity greater than 0.5. To measure the similarity between classes we average the embeddings for all instances of a class, reducing it to pair-wise relationships. The figure shows that we are better at organizing concepts hierarchically, which as a consequence means that inputs with hierarchically similar concepts are more likely to fall in the same quantization bin. This enables better hierarchical performance even at low bit length. We suspect this is because spherical embeddings are learned by forcing classes to be equally dissimilar, whereas in hyperbolic space we can enforce a margin between classes while keeping track of siblings due to its infinite boundary nature.

### 4.5 EFFECT OF CURVATURE AND RADIUS

The curvature of the Poincaré disk model is determined by $c$; $r$ controls the radius for constructing prototypes (optionally with the hierarchy $\mathcal{H}$) $\boldsymbol{p}_i$, with which we learn $f(\cdot)$ to map the input samples

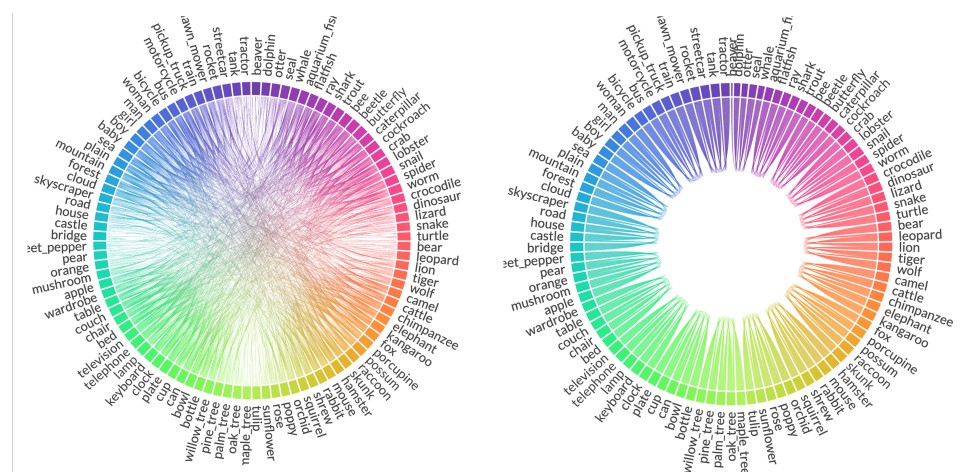

Figure 5: **Visualization of pair-wise class similarities** for (left) non-hierarchical spherical embeddings and (right) our hierarchical binary embeddings. Our embeddings are organized more hierarchically, balancing strong quantization and hierarchical performance.

Table 4: **The effect of curvature** on CIFAR100. Parameter $r^2$ is the squared radius used when constructing the class prototypes $\boldsymbol{P}$ on poincaré disk, whereas $c$ is the curvature used when optimizing $f(\cdot)$ to map data samples to class prototypes. Low curvature indicates nearly uncurved, Euclidean-like space, where high curvature causes numerical instability. Hence a middle ground for class prototype embedding and sample embedding is preferred.

| | $r^2 = 10^3$ | $r^2 = 10^2$ | $r^2 = 10$ | $r^2 = 1$ | $r^2 = 0.1$ |
|---|---|---|---|---|---|
| $c = 10^{-3}$ | 0.7045 ±0.0027 | 0.7294 ±0.0021 | 0.7930 ±0.0029 | 0.7810 ±0.0031 | 0.7921 ±0.0018 |
| $c = 10^{-2}$ | - | 0.7889 ±0.0024 | 0.7939 ±0.0029 | 0.7947 ±0.0025 | 0.7896 ±0.0019 |
| $c = 0.1$ | - | - | 0.7938 ±0.0030 | **0.7948** ±0.0023 | 0.7938 ±0.0021 |
| $c = 1$ | - | - | - | 0.7463 ±0.0015 | 0.7817 ±0.0020 |
| $c = 10$ | - | - | - | - | 0.5627 ±0.0013 |

to the class prototypes. Meanwhile, $c$ is the curvature of $\mathbb{D}$ where we embed images and videos, it can be regarded as adjusting the hyperbolic metric, resulting in a different distance calculation with the same prototypes. In Table 4 we compare different settings for $c$ and $r$ and find an interaction between the two parameters, with the highest performance obtained with a high $r$ and a low $c$. Overall, training class prototypes with an intermediate curvature is preferred, we suspect that is because the class prototypes are not pushed to the disk boundary, thereby leaving some room for embedding class instances in the later stage. More analyses of other datasets are given in Appendix D.3.

## 4.6 PRODUCT QUANTIZATION AND LARGE-SCALE COMPARISONS

Both our method and classical product quantization can be regarded as post-processing on top of existing feature vectors. We draw a comparison in Table 5. Besides its superior performance, we find, interestingly, if we combine our method with PQ, i.e., using the binary hyperbolic embeddings as the feature vectors for PQ, it further improves the performance of PQ. This might be because our binary grouping in eq equation 8 already groups each group of sub-space, thus better aligned with PQ's subspace division. We furthermore compare to hashing methods, which require a long training time and are not easily scalable to large datasets. As a result, recent hashing research has focused on small datasets for validation. Therefore, in this paper, due to the outsized training cost, we do not compare the performance of hashing methods on large datasets. In Table 6, we show that on CIFAR-100, our approach is on par with the hashing methods, while we do not require complex optimization and our performance does not decrease after as bit-length goes beyond 128.

Table 5: **Comparison and combination with product quantization.** Product quantization is a canonical approach in retrieval, but not competitive to our binary hyperbolic embeddings in either the unsupervised or the supervised setting.

| | CIFAR100 | | | ImageNet1K-Val | | |
| | 128bit | 256bit | 512bit | 128bit | 256bit | 512bit |
|---|---|---|---|---|---|---|
| Unsupervised PQ (Jégou et al., 2011) | $0.5213 \pm 0.0009$ | $0.5372 \pm 0.0012$ | $0.5503 \pm 0.0015$ | $0.3574 \pm 0.0016$ | $0.4152 \pm 0.0018$ | $0.4573 \pm 0.0021$ |
| Unsupervised Ours | $0.5304 \pm 0.0011$ | $0.5412 \pm 0.0013$ | $0.5497 \pm 0.0016$ | $0.3736 \pm 0.0015$ | $0.4213 \pm 0.0019$ | $0.4584 \pm 0.0020$ |
| $\mathbb{D}$ + PQ (Jégou et al., 2011) | $0.7102 \pm 0.0019$ | $0.7304 \pm 0.0022$ | $0.7353 \pm 0.0025$ | $0.5452 \pm 0.0029$ | $0.6051 \pm 0.0030$ | $0.6082 \pm 0.0031$ |
| $\mathbb{D}$ + OPQ (Ge et al., 2014) | $0.7613 \pm 0.0018$ | $0.7714 \pm 0.0020$ | $0.7782 \pm 0.0021$ | $0.5681 \pm 0.0025$ | $0.6212 \pm 0.0032$ | $0.6213 \pm 0.0030$ |
| Ours + OPQ | $0.7747 \pm 0.0016$ | $0.7726 \pm 0.0012$ | $0.7814 \pm 0.0026$ | $0.5701 \pm 0.0022$ | $0.6223 \pm 0.0028$ | $0.6253 \pm 0.0033$ |
| Ours | $\mathbf{0.8031} \pm 0.0010$ | $\mathbf{0.8090} \pm 0.0040$ | $\mathbf{0.8177} \pm 0.0030$ | $\mathbf{0.5814} \pm 0.0104$ | $\mathbf{0.6365} \pm 0.0067$ | $\mathbf{0.6358} \pm 0.0006$ |

Table 6: **Comparison to hashing methods** on CIFAR-100 Our approach is preferred, without incurring complex optimization, while the performance kept increasing without being limited to a high number of bits.

| | 16bits | 32bits | 64bits | 128bits | 512bits |
|---|---|---|---|---|---|
| CSQ (Yuan et al., 2020) | $0.6473 \pm 0.003$ | $\mathbf{0.7562} \pm 0.002$ | $\mathbf{0.7981} \pm 0.001$ | $0.8101 \pm 0.002$ | $0.8072 \pm 0.002$ |
| DPN (Fan et al., 2020) | $0.6370 \pm 0.004$ | $0.7431 \pm 0.003$ | $0.7902 \pm 0.002$ | $0.8090 \pm 0.002$ | $0.8130 \pm 0.003$ |
| OrthoCos (Hoe et al., 2021a) | $0.6630 \pm 0.004$ | $0.7552 \pm 0.003$ | $0.7905 \pm 0.003$ | $\mathbf{0.8120} \pm 0.002$ | $0.8012 \pm 0.002$ |
| BiHalf (Li & van Gemert, 2021) | $0.6775 \pm 0.003$ | $0.7572 \pm 0.002$ | $0.7887 \pm 0.002$ | $0.7977 \pm 0.001$ | $0.7803 \pm 0.002$ |
| Hyperbolic-Hashing (Yu et al., 2024) | $0.2490 \pm 0.012$ | $0.3921 \pm 0.009$ | $0.4893 \pm 0.008$ | $0.5693 \pm 0.007$ | $0.6013 \pm 0.002$ |
| Ours | $0.5370 \pm 0.002$ | $0.6990 \pm 0.001$ | $0.7706 \pm 0.005$ | $0.8031 \pm 0.001$ | $\mathbf{0.8177} \pm 0.003$ |

To showcase the potential of our approach on large-scale settings, we also experiment on the Quick Draw dataset (Google, 2023). Here, we embed the raw 50 Million images with a simple MLP backbone followed by a Euclidean or hyperbolic head. To avoid out-of-memory issues, we use 64 embedding dimensions, and 4-bit quantization and perform retrieval on two randomly picked query sets (one small scale and one large scale) of the test set. The results in Table 7 show that on such large-scale settings, binary hyperbolic embeddings remain highly effective.

Table 7: **Large-scale evaluation on QuickDraw**, binary hyperbolic embeddings are scalable as well.

| | **QuickDraw-50K** | **QuickDraw-10M** |
|---|---|---|
| Euclidean Binarization | 0.2445 | 0.0407 |
| Ours | **0.3149** | **0.0712** |

## 5 CONCLUSION

Hyperbolic deep learning has a wide range of applications, from images to videos. However, its application in large-scale search has been hampered by slow distance calculations. In this work, we overcome this limitation by proving the retrieval equivalence between hyperbolic and Hamming distances, which allows binarization of the hyperbolic space and significantly accelerates distance calculations. We experimentally verify this acceleration, across the video and image domain, obtaining significant accerlation at roughly equal performance. Our hyperbolic binary embeddings demonstrate the viability of hyperbolic space for large-scale retrieval and hierarchical retrieval.

**Broader impact.** The proposed model, through binarization of hyperbolic space, substantially decreases memory consumption and computational costs for retrieval, which contributes to reducing energy costs and infrastructure expenses.

**Limitations.** A limitation of hyperbolic embeddings on a Poincaré disk is the issue of numerical stability (Yu & De Sa, 2019; Mishne et al., 2023), as it relies on a finite numerical range to represent an infinite volume. This constraint becomes particularly pronounced because the Poincaré disk model compresses distances exponentially as they approach the boundary, leading to substantial precision challenges. An induced limitation is that we binarize the embeddings uniformly in a non-uniform space, which would be an inspiration for future work. Double-precision arithmetic is recommended to mitigate numerical instabilities near the disk's edge.

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

## A  HYPERBOLIC MACHINE LEARNING PRELIMINARIES

Hyperbolic geometry is non-Euclidean geometry characterized by a constant negative curvature. Unlike Euclidean geometry, hyperbolic space exhibits unique properties, such as the exponential growth of volume (Ganea et al., 2018b) with respect to the radius and the divergence of parallel lines. It serves as a powerful mathematical framework for modeling hierarchical data, tree-like structures, and other settings where distances grow exponentially.

Hyperbolic spaces are equipped with several equivalent models  Cannon et al. (1997) that offer different perspectives on the same underlying geometry. These models provide flexibility for computation, visualization, and mathematical reasoning. Despite their different formulations, they are isometric, meaning their distance metrics can be transformed into one another through well-defined coordinate transformations.

### A.1  FIVE COMMON MODELS OF HYPERBOLIC GEOMETRY

Following the conventions in differential geometry, the five commonly used models of hyperbolic geometry are isometric to each other:

**1. Halfspace Model (H):**  Also known as the Poincaré halfspace model, it is defined as:

$$H = \{(1, x_2, \ldots, x_{n+1}) : x_{n+1} > 0\}.$$

In this model, hyperbolic space is represented within a half-plane or half-space above a given axis, often used in complex analysis and conformal mapping.

**2. Interior of the Disk Model (I):**   Also known as the Poincaré disk model, it is given by:

$$I = \left\{ (x_1, \ldots, x_n, 0) : x_1^2 + \cdots + x_n^2 < 1 \right\}.$$

Here, the hyperbolic space resides inside a Euclidean unit disk. This model is conformal, preserving angles, and is commonly used in visualization.

**3. Jemisphere Model (J):**   Pronounced with the "J" as in Spanish, this model is defined as:

$$J = \left\{ (x_1, \ldots, x_{n+1}) : x_1^2 + \cdots + x_{n+1}^2 = 1 \text{ and } x_{n+1} > 0 \right\}.$$

The hyperbolic space is represented as a hemisphere of a unit sphere.

**4. Klein Model (K):**   This model, known for its projective properties, is given by:

$$K = \left\{ (x_1, \ldots, x_n, 1) : x_1^2 + \cdots + x_n^2 < 1 \right\}.$$

The Klein model preserves straight-line geodesics, making it useful for certain computations, though it does not preserve angles.

**5. Loid Model (L):**   Short for the hyperboloid model, it is defined as:

$$L = \left\{ (x_1, \ldots, x_n, x_{n+1}) : x_1^2 + \cdots + x_n^2 - x_{n+1}^2 = -1 \text{ and } x_{n+1} > 0 \right\}.$$

This model is particularly important in physics and mathematics due to its direct connection to Lorentzian geometry and relativity.

A.2   Isometry Between Models

The equivalence of these models is established through isometric coordinate transformations, as detailed in Section 7 of Cannon et al. (1997). These transformations ensure that distances and geodesics remain invariant when transitioning between models, making them interchangeable depending on the application.

**Remark**   Under the isometry defined in Cannon et al. (1997), Section 7, five hyperbolic models yield equivalent retrieval results.

*Proof.*  Although we do not binarize other hyperbolic models, they can still use our binarization via isometry defined in hyperbolic geometry Cannon et al. (1997), with a single coordinates transformation. In particular, we give one example of how the Lorentz model $\mathcal{L}$ transforms to Poincaré disk model Radford et al. (2021) in one line:

$$\mathcal{L} \mapsto \mathbb{D} : (x_0, x_1, ..., x_n) \mapsto \left( \frac{x_1}{1 + x_0}, \frac{x_2}{1 + x_0}, ..., \frac{x_n}{1 + x_0} \right) \tag{12}$$

We similarly illustrate the other isometries following the convention in (Cannon et al., 1997):

From Lorentz model to Klein model:

$$(x_0, x_1, \cdots, x_n) \mapsto (x_1/x_0, x_2/x_0, \cdots, x_n/x_0). \tag{13}$$

From Lorentz model to Hemisphere model:

$$(x_0, x_1, \cdots, x_n) \mapsto (x_0/x_n, x_1/x_n, \cdots, x_{n-1}/x_n, 1/x_n). \tag{14}$$

From Hemisphere model to Poincaé Halfspace model:

$$(x_0, x_1, \cdots, x_n) \mapsto (1, 2x_1/(x_0 + 1), \cdots, 2x_n/(x_0 + 1)) \tag{15}$$

From Hemisphere model to Poincaré Disk model:

$$(x_0, x_1, \cdots, x_n) \mapsto (x_0/(x_n + 1), x_1/(x_n + 1), \cdots, x_{n-1}/(x_n + 1)) \tag{16}$$

Note that all the above transformations are isometry, meaning that the distance between source model embeddings is equal to the distance between the target model embeddings. Therefore, any hyperbolic model can benefit from our Poincaré binarization. □

### A.3 POINCARÉ BALL GEOMETRY

Hyperbolic geometry is a non-Euclidean geometry characterized by a constant negative curvature. Among the five commonly used models of hyperbolic geometry, we choose the Poincaré ball model due to its unique properties of isotropy and symmetry in each dimension. These properties make the Poincaré ball particularly well-suited for designing simple and effective binarization strategies.

The Poincaré ball model is defined as:

$$\mathbb{D}^n = \{\mathbf{x} \in \mathbb{R}^n : \|\mathbf{x}\| < 1\}, \tag{17}$$

where $\|\cdot\|$ denotes the Euclidean norm. Within this model, the geometry is described by a conformal Riemannian metric that preserves angles, making it especially convenient for embedding hierarchical or tree-like data structures.

The Riemannian metric of the Poincaré ball is given by:

$$g_{\mathbf{x}} = \lambda_{\mathbf{x}}^2 g^E, \tag{18}$$

where $g^E$ is the Euclidean metric and $\lambda_{\mathbf{x}} = \frac{2}{1-\|\mathbf{x}\|^2}$ is the conformal factor. This conformal property ensures that local distances and directions are geometrically meaningful, while the ball's bounded nature simplifies computations and facilitates compact embedding representations.

### A.4 DISTANCE ON POINCARÉ BALL

Hyperbolic space can have different constant negative curvatures, parameterized by $c$. For simplicity, we follow the conventions in Ganea et al. (2018a); Shimizu et al. (2021) that use $c > 0$ to represent negative curvature to simplify computation. The Poincaré ball model with curvature $c$ is defined as:

$$\mathbb{D}_c^n = \{\mathbf{x} \in \mathbb{R}^n : \|\mathbf{x}\| < \frac{1}{\sqrt{c}}\}, \tag{19}$$

where $\|\cdot\|$ denotes the Euclidean norm. The hyperbolic distance between two points $\mathbf{x}, \mathbf{y} \in \mathbb{D}_c^n$ under curvature $c$ is given by:

$$d_c(\mathbf{x}, \mathbf{y}) = \frac{2}{\sqrt{c}} \tanh^{-1} \left( \sqrt{c} \|\ominus_c \mathbf{x} \oplus_c \mathbf{y}\| \right). \tag{20}$$

where the Möbius addition $\oplus_c$ and subtraction $\ominus_c$ are defined as:

$$\mathbf{x} \oplus_c \mathbf{y} = \frac{\left(1 + 2c\langle\mathbf{x}, \mathbf{y}\rangle + c\|\mathbf{y}\|^2\right)\mathbf{x} + \left(1 - c\|\mathbf{x}\|^2\right)\mathbf{y}}{1 + 2c\langle\mathbf{x}, \mathbf{y}\rangle + c^2\|\mathbf{x}\|^2\|\mathbf{y}\|^2}, \tag{21}$$

$$\mathbf{x} \ominus_c \mathbf{y} = \mathbf{x} \oplus_c (-\mathbf{y}), \tag{22}$$

$$\ominus_c \mathbf{x} = \mathbf{0} \oplus_c (-\mathbf{x}), \tag{23}$$

When $c = -1$, this reduces to the standard Poincaré ball distance:

$$d_{\mathbb{D}}(\mathbf{x}, \mathbf{y}) = \operatorname{arcosh} \left( 1 + 2\frac{\|\mathbf{x} - \mathbf{y}\|^2}{(1 - \|\mathbf{x}\|^2)(1 - \|\mathbf{y}\|^2)} \right). \tag{24}$$

Here, $c$ allows for the scaling of the hyperbolic space, with smaller $|c|$ corresponding to a "flatter" space.

#### A.4.1 RECOVERING EUCLIDEAN SPACE AS $c \to 0$

As the curvature $c$ approaches 0 (i.e., $c \to 0$), the hyperbolic space becomes increasingly flat, and its geometry converges to Euclidean geometry. Specifically:

- The radius of the Poincaré ball, $\frac{1}{\sqrt{c}}$, tends to infinity.

- The hyperbolic distance formula simplifies to the Euclidean distance:

$$\lim_{c \to 0^-} d_{\mathbb{D}_c}(\mathbf{x}, \mathbf{y}) = \|\mathbf{x} - \mathbf{y}\|. \tag{25}$$

- This occurs also because the conformal factor $\lambda_{\mathbf{x}} = \frac{2}{1-c\|\mathbf{x}\|^2}$ in the Riemannian metric approaches 1, and the distortion due to curvature vanishes.,..

## A.5 POINCARÉ DISTANCE SYMMETRY UNDER CURVATURE $c$

We show that $d_c(\mathbf{x}, \mathbf{y})$ is symmetric, despite the non-commutative nature of the Möbius addition $\oplus_c$. This symmetry arises from the fact that the Euclidean norm of the Möbius addition is commutative. Specifically, the norm of the Möbius addition is given by:

$$\|\mathbf{x} \oplus_c \mathbf{y}\| = \sqrt{\frac{\|\mathbf{x}\|^2 + 2\langle \mathbf{x}, \mathbf{y}\rangle + \|\mathbf{y}\|^2}{1 + 2c\langle \mathbf{x}, \mathbf{y}\rangle + c^2\|\mathbf{x}\|^2\|\mathbf{y}\|^2}}. \tag{26}$$

The key observation here is that while $\mathbf{x} \oplus_c \mathbf{y} \neq \mathbf{y} \oplus_c \mathbf{x}$ in general, the norm satisfies:

$$\|\mathbf{x} \oplus_c \mathbf{y}\| = \|\mathbf{y} \oplus_c \mathbf{x}\|, \tag{27}$$

due to the symmetric structure of the numerator and denominator in the norm formula.

As a result, the hyperbolic distance $d_c(\mathbf{x}, \mathbf{y})$ becomes symmetric:

$$d_c(\mathbf{x}, \mathbf{y}) = d_c(\mathbf{y}, \mathbf{x}). \tag{28}$$

This property is crucial for defining a proper distance metric in hyperbolic space under curvature $c$.

## A.6 COMPLEXITY OF DISTANCE $d_{\mathbb{D}}(\mathbf{q}, \mathbf{x})$:

$$d_{\mathbb{D}}(\mathbf{q}, \mathbf{x}) = \frac{2}{\sqrt{c}} \tanh^{-1}\left(\sqrt{c} \left\| \frac{(1 - 2c\langle \mathbf{q}, \mathbf{x}\rangle + c\|\mathbf{x}\|^2)\mathbf{q} + (1 - c\|\mathbf{q}\|^2)\mathbf{x}}{1 - 2c\langle \mathbf{q}, \mathbf{x}\rangle + c^2\|\mathbf{q}\|^2\|\mathbf{x}\|^2} \right\|^2\right) \tag{29}$$

ANALYSIS OF OPERATIONS

- Inner products and norms:
    - $\langle \mathbf{q}, \mathbf{x}\rangle$: $n$ multiplications, $n - 1$ additions
    - $\|\mathbf{q}\|^2$ and $\|\mathbf{x}\|^2$: $2n$ multiplications, $2(n-1)$ additions
- Scalar computations:
    - Few multiplications and additions (constants)
- Vector operations:
    - Scaling vectors: $2n$ multiplications
    - Vector addition: $n$ additions
- Norm squared of numerator:
    - $n$ multiplications, $n - 1$ additions
- Division & Transcendental function:
    - One division (scalar)
    - One $\tanh^{-1}$ evaluation
- Total Operations:
    - Multiplications: Approximately $7n$ + constants
    - Additions/Subtractions: Approximately $5n$ + constants
    - Divisions & Transcendental function: One

CONCLUSION:

- While both distances have linear complexity $O(n)$, $d_{\mathbb{D}}(\mathbf{q}, \mathbf{x})$ involves significantly more arithmetic operations.
- Therefore, even when considering constant factors, $d_{\mathbb{D}}(\mathbf{q}, \mathbf{x})$ is computationally more intensive than $d_{\mathbb{R}}(\mathbf{x}, \mathbf{y})$.

## B    REAL-VALUED HYPERBOLIC EMBEDDINGS

Given a training set $\mathcal{D}_{\text{train}} = \{(x_i, y_i)\}_{i=1}^N$. For each class $y$, let $\boldsymbol{p}_y$ denote its prototype target in a hyperbolic embedding space. We aim to train a network $f(\cdot)$ that projects the data point $x$ onto hyperbolic space $\mathbf{x} = f(x)$, where $f(\cdot)$ is an arbitrary backbone network with an exponential map on top to project the output embedding to hyperbolic space. The likelihood of sample $(x, y)$ is given as:

$$p(x|y = y') = \frac{\exp\left(-\mathrm{d}_{\mathbb{D}}(f(x), \boldsymbol{p}_{y'})\right)}{\sum_{y''}^{|\mathcal{H}|} \exp\left(-\mathrm{d}_{\mathbb{D}}(f(x), \boldsymbol{p}_{y''})\right)}, \tag{30}$$

which is optimized through the negative log-likelihood loss akin to Long et al. (2020). For the prototype embeddings of the classes, our approach works both in the one-vs-rest setting and the setting where we are equipped with label hierarchy $\mathcal{H}$. In the one-vs-rest setting, we can simply define class prototypes as maximally separated prototypes akin to Kasarla et al. (2022). In the hierarchical setting, we can obtain class prototypes for example using Hyperbolic Entailment Cones Ganea et al. (2018b).

## C    PROOFS

**Proposition C.1.** *Given query $\mathbf{q}$, For any $\mathbf{x}_1, \mathbf{x}_2$ in the database such that $|\|\mathbf{x}_1\| - \|\mathbf{x}_2\|| = \epsilon$, given the Poincar'e disk boundary tolerance margin $\delta$, if $0 \le \epsilon \le \delta \frac{\mathrm{d}_{\mathbb{R}}(\mathbf{q}, \mathbf{x}_2) - \mathrm{d}_{\mathbb{R}}(\mathbf{q}, \mathbf{x}_1)}{\mathrm{d}_{\mathbb{R}}(\mathbf{q}, \mathbf{x}_2)}$, then $\mathrm{d}_{\mathbb{R}}(\mathbf{q}, \mathbf{x}_1) \le \mathrm{d}_{\mathbb{R}}(\mathbf{q}, \mathbf{x}_2)$ implies $\mathrm{d}_{\mathbb{D}}(\mathbf{q}, \mathbf{x}_1) \le \mathrm{d}_{\mathbb{D}}(\mathbf{q}, \mathbf{x}_2)$.*

*Proof.* we start the analysis from the fact that poincar'e disk usually defines a tolerance margin $\delta$ such that

$$\|x\|^2 \le 1 - \delta \tag{31}$$

to avoid numerical instability. For example, Khrulkov et al. (2020) adopted $\delta = 10^{-3}$ as the margin. we show that 1) preserving $\mathrm{d}_{\mathbb{D}}(q, x_1) < \mathrm{d}_{\mathbb{D}}(q, x_2)$ is equivalent to $r \in (L, R)$ for some $L, R$.

Note that $\mathrm{d}_{\mathbb{D}}(q, x_1) < \mathrm{d}_{\mathbb{D}}(q, x_2)$ is equivalent to

$$\cosh^{-1}\left(1 + 2\frac{\|q - x_1\|^2}{(1 - \|q\|^2)(1 - \|x_1\|^2)}\right) < \cosh^{-1}\left(1 + 2\frac{\|q - x_2\|^2}{(1 - \|q\|^2)(1 - \|x_2\|^2)}\right). \tag{32}$$

As $\cosh^{-1}(\cdot)$ is strictly increasing, $\mathrm{d}_{\mathbb{D}}(q, x_1) < \mathrm{d}_{\mathbb{D}}(q, x_2)$ is equivalent to

$$\frac{\|q - x_1\|^2}{(1 - \|q\|^2)(1 - \|x_1\|^2)} < \frac{\|q - x_2\|^2}{(1 - \|q\|^2)(1 - \|x_2\|^2)}, \tag{33}$$

which is equivalent to

$$(\|q - x_1\|^2)(1 - \|x_2\|^2) < (\|q - x_2\|^2)(1 - \|x_1\|^2). \tag{34}$$

With simplified notation as:

$$\mathrm{d}_{\mathbb{R}}(q, x_1)^2(1 - r^2) < \mathrm{d}_{\mathbb{R}}(q, x_2)^2(1 - (r \pm \epsilon)^2). \tag{35}$$

Solving this quadratic inequality with respect to $r$, we have $r \in (L, R)$, where

$$L = \frac{\pm\epsilon \mathrm{d}_{\mathbb{R}}(q, x_2) - \sqrt{(\mathrm{d}_{\mathbb{R}}(q, x_2) - \mathrm{d}_{\mathbb{R}}(q, x_1))^2 + \epsilon^2(2\mathrm{d}_{\mathbb{R}}(q, x_2)^2 - \mathrm{d}_{\mathbb{R}}(q, x_1))}}{\mathrm{d}_{\mathbb{R}}(q, x_2) - \mathrm{d}_{\mathbb{R}}(q, x_1)}, \tag{36}$$

$$R = \frac{\pm\epsilon \mathrm{d}_{\mathbb{R}}(q, x_2) + \sqrt{(\mathrm{d}_{\mathbb{R}}(q, x_2) - \mathrm{d}_{\mathbb{R}}(q, x_1))^2 + \epsilon^2(2\mathrm{d}_{\mathbb{R}}(q, x_2)^2 - \mathrm{d}_{\mathbb{R}}(q, x_1))}}{\mathrm{d}_{\mathbb{R}}(q, x_2) - \mathrm{d}_{\mathbb{R}}(q, x_1)}, \tag{37}$$

leading to the following inequalities:

$$L < \frac{\pm \epsilon \mathrm{d}_{\mathbb{R}}(q, x_2) - \sqrt{(\mathrm{d}_{\mathbb{R}}(q, x_2) - \mathrm{d}_{\mathbb{R}}(q, x_1))^2}}{\mathrm{d}_{\mathbb{R}}(q, x_2) - \mathrm{d}_{\mathbb{R}}(q, x_1)} \tag{38}$$

$$\leq \frac{\epsilon \mathrm{d}_{\mathbb{R}}(q, x_2) - \sqrt{(\mathrm{d}_{\mathbb{R}}(q, x_2) - \mathrm{d}_{\mathbb{R}}(q, x_1))^2}}{\mathrm{d}_{\mathbb{R}}(q, x_2) - \mathrm{d}_{\mathbb{R}}(q, x_1)} = \epsilon \frac{\mathrm{d}_{\mathbb{R}}(q, x_2)}{\mathrm{d}_{\mathbb{R}}(q, x_2) - \mathrm{d}_{\mathbb{R}}(q, x_1)} - 1 \tag{39}$$

$$R > \frac{\pm \epsilon \mathrm{d}_{\mathbb{R}}(q, x_2) + \sqrt{(\mathrm{d}_{\mathbb{R}}(q, x_2) - \mathrm{d}_{\mathbb{R}}(q, x_1))^2}}{\mathrm{d}_{\mathbb{R}}(q, x_2) - \mathrm{d}_{\mathbb{R}}(q, x_1)} \tag{40}$$

$$\geq \frac{-\epsilon \mathrm{d}_{\mathbb{R}}(q, x_2) + \sqrt{(\mathrm{d}_{\mathbb{R}}(q, x_2) - \mathrm{d}_{\mathbb{R}}(q, x_1))^2}}{\mathrm{d}_{\mathbb{R}}(q, x_2) - \mathrm{d}_{\mathbb{R}}(q, x_1)} = -\epsilon \frac{\mathrm{d}_{\mathbb{R}}(q, x_2)}{\mathrm{d}_{\mathbb{R}}(q, x_2) - \mathrm{d}_{\mathbb{R}}(q, x_1)} + 1, \tag{41}$$

which means the quadratic inequality $\mathrm{d}_{\mathbb{R}}(q, x_1)^2(1 - r^2) < \mathrm{d}_{\mathbb{R}}(q, x_2)^2(1 - (r \pm \epsilon)^2)$ holds true when

$$\epsilon \frac{\mathrm{d}_{\mathbb{R}}(q, x_2)}{\mathrm{d}_{\mathbb{R}}(q, x_2) - \mathrm{d}_{\mathbb{R}}(q, x_1)} - 1 \leq -1 + \delta \tag{42}$$

$$-\epsilon \frac{\mathrm{d}_{\mathbb{R}}(q, x_2)}{\mathrm{d}_{\mathbb{R}}(q, x_2) - \mathrm{d}_{\mathbb{R}}(q, x_1)} + 1 \leq 1 - \delta \tag{43}$$

holds, leading that

$$\epsilon \leq \delta \frac{\mathrm{d}_{\mathbb{R}}(q, x_2) - \mathrm{d}_{\mathbb{R}}(q, x_1)}{\mathrm{d}_{\mathbb{R}}(q, x_2)}$$

Therefore, we conclude that the upper bound of $\epsilon$ is $\epsilon \leq \delta \frac{\mathrm{d}_{\mathbb{R}}(q,x_2)-\mathrm{d}_{\mathbb{R}}(q,x_1)}{\mathrm{d}_{\mathbb{R}}(q,x_2)} = \frac{\mathrm{d}_{\mathbb{R}}(q,x_2)-\mathrm{d}_{\mathbb{R}}(q,x_1)}{\mathrm{d}_{\mathbb{R}}(q,x_2)}$. We also have a trivial lower bound of $\epsilon$, as $\epsilon \geq 0$, this is because when $\epsilon = 0$, we easily have $L = -1, R = 1$, are the boundary of the Poincare disk, leading to a **constantly true proposition**. $\qquad \square$

**Proposition C.2.** *(Binary Ranking Preservation) For a binarizer $g(\cdot)$ such that $\langle \mathbf{x}^+, \mathbf{y}^+ \rangle \propto \langle g(\mathbf{x}), g(\mathbf{y}) \rangle$, $\mathrm{d}_{\mathbb{D}}(\mathbf{x}, \mathbf{y})$ is yields the same ranking results to Hamming distance $\mathrm{d}_{\mathbb{H}}(\mathbf{x}^b, \mathbf{y}^b) = \|\mathbf{x}^b \oplus \mathbf{y}^b\|_0$ for nearest neighbor search.*

*Proof.* For any binary representation $\mathbf{x}^b = g(\mathbf{x}), \mathbf{y}^b = g(\mathbf{y})$, the squared Euclidean distance $\mathrm{d}_{\mathbb{R}}$ is proportional to Hamming distance $\mathrm{d}_{\mathbb{H}}$:

$$\begin{aligned} d_{\mathbb{R}}^2(\mathbf{x}, \mathbf{y}) = d_{\mathbb{R}}^2(\mathbf{x}^+, \mathbf{y}^+) &= \|\mathbf{x}^+ - \mathbf{y}^+\|^2 \\ &= \|\mathbf{x}^+\|^2 + \|\mathbf{y}^+\|^2 - 2\langle \mathbf{x}^+, \mathbf{y}^+ \rangle \\ &\propto \|g(\mathbf{x})\|^2 + \|g(\mathbf{y})\|^2 - 2\langle g(\mathbf{x}), g(\mathbf{y}) \rangle \\ &= \|\mathbf{x}^b\|^2 + \|\mathbf{y}^b\|^2 - 2\langle \mathbf{x}^b, \mathbf{y}^b \rangle \\ &= \mathrm{d}_{\mathbb{H}}(\mathbf{x}^b, \mathbf{y}^b). \end{aligned} \tag{44}$$

Following Proposition C.2, we have nearest neighbor equivalence between the Hamming distance and hyperbolic distance:

$$\begin{aligned} \arg\min_{\mathbf{x}} \mathrm{d}_{\mathbb{H}}(\mathbf{q}, \mathbf{x}) &= \arg\min_{\mathbf{x}} d_{\mathbb{R}}^2(\mathbf{q}, \mathbf{x}) \quad \text{by equation 44} \\ &= \arg\min_{\mathbf{x}} \mathrm{d}_{\mathbb{R}}(\mathbf{q}, \mathbf{x}) \\ &= \arg\min_{\mathbf{x}} \mathrm{d}_{\mathbb{D}}(\mathbf{q}, \mathbf{x}) \quad \text{by Prop. C.1.} \end{aligned} \tag{45}$$

$\qquad \square$

**Corollary C.3.** *Under the condition of Proposition C.2, hyperbolic metric $d_{\mathbb{D}}(\mathbf{x}, \mathbf{y})$ is ranking preserving as euclidean metric $\mathrm{d}_{\mathbb{R}}(\mathbf{x}, \mathbf{y})$.*

*Proof.* Given query $\mathbf{q}$ and database $\mathcal{X} = \{\mathbf{x}_i\}$, we annotate the subscript of the database according to the Euclidean distance to the query. such that:

$$\mathrm{d}_{\mathbb{R}}(\mathbf{q}, \mathbf{x}_1) \leq \mathrm{d}_{\mathbb{R}}(\mathbf{q}, \mathbf{x}_2) \leq \cdots \leq \mathrm{d}_{\mathbb{R}}(\mathbf{q}, \mathbf{x}_N) \tag{46}$$

From Proposition C.2, we can have:

$$d_{\mathbb{D}}(\mathbf{q}, \mathbf{x}_1) \leq d_{\mathbb{D}}(\mathbf{q}, \mathbf{x}_2) \leq \cdots \leq d_{\mathbb{D}}(\mathbf{q}, \mathbf{x}_N) \tag{47}$$

$$\square$$

This means under the conditions of Proposition C.2 the hyperbolic metric yields the same output ranking as using the Euclidean metric, leading to the same retrieval outputs.

## D  SLOWNESS OF HYPERBOLIC SPACE

The strongest evidence that hyperbolic space is more suitable for retrieval than Euclidean is its **compact nature**, as shown in (Long et al., 2020) and Ermolov et al. (2022). Unfortunately, this comes at the cost of **slowness** that is determined by the nature of the hyperbolic distance:

$$d_{\mathbb{D}}(\mathbf{q}, \mathbf{x}) = \frac{2}{\sqrt{c}} \tanh^{-1} \left( \sqrt{c} \left\| \frac{\left(1 - 2c\langle \mathbf{q}, \mathbf{x} \rangle + c\|\mathbf{x}\|^2\right) \mathbf{q} + \left(1 - c\|\mathbf{q}\|^2\right) \mathbf{x}}{1 - 2c\langle \mathbf{q}, \mathbf{x} \rangle + c^2\|\mathbf{q}\|^2\|\mathbf{x}\|^2} \right\|^2 \right). \tag{48}$$

However, The efficiency brought by low-dimensional embedding is offset by its slow computation, in hyperbolic space, distance calculation is much slower than its Euclidean counterpart

$$d_{\mathbb{R}}(\mathbf{x}, \mathbf{y}) = \|\mathbf{x} - \mathbf{y}\|^2. \tag{49}$$

We can quantitatively show how slow hyperbolic distance calculation is, as hyperbolic distance calculation takes $20.2713 \pm 0.1443$ micro-seconds, while its Euclidean counterpart takes $4.7033 \pm 0.0394$ micro-seconds. This result comes from 512-dim float number vectors distance calculations, averaged over $10^4$ calculations on a single CPU core, both $d_{\mathbb{D}}$ and $d_{\mathbb{R}}$ are based on implementation of (Becigneul & Ganea, 2019).

To make hyperbolic space practically useful for retrieval, we need to bypass the need for explicit distance calculations in hyperbolic space. We present the necessary theory and algorithms to do so in our Binary Hyperbolic Embeddings.

### D.1  ANALYSIS FOR LORENTZ MODEL

The distance metric under Lorentz model is induced by *Lorentzian scalar product*:

$$\langle \mathbf{x}, \mathbf{y} \rangle_{\mathcal{L}} = x_0 y_0 + \sum_{i=1}^{n} x_i y_i \tag{50}$$

The associated distance is defined as:

$$d_{\mathbb{L}}(\mathbf{x}, \mathbf{y}) = cosh^{-1}\left(-\langle \mathbf{x}, \mathbf{y} \rangle_{\mathcal{L}}\right) \tag{51}$$

Although this distance appears to be fast to compute, we still suffer from the complexity of its anisotropic and unlimited range. Using Poincaré disk as a proxy also does not work, because transforming from Poincaré disk $\mathbb{D}$ to Lorentz model $\mathcal{L}$ also incurs high computation costs:

$$\mathbb{D} \mapsto \mathcal{L} : (\mathbf{x}) \mapsto \frac{(1 + \|\mathbf{x}\|^2, x_1, x_2, \cdots, x_n)}{1 - \|\mathbf{x}\|^2} \tag{52}$$

## E  MORE EXPERIMENTS

### E.1  EFFECT OF DIMENSIONALITY AND NUMBER OF BITS

We visually show the speed-performance trade-off on CIFAR100 in Figure E.1. Lower bit/dimension configurations result in higher speeds (up to 62.73×) but lower mAP, while higher configurations provide better mAP (up to 0.745) at the cost of reduced speed (down to 4.71×). The best trade-off between speed and performance appears to be with binary hyperbolic embeddings, achieving over 8 times faster speeds with roughly the same performance. The highest mAP of 0.745 is achieved with both $128 \times 2 = 256$ and $128 \times 4 = 512$ configurations, with the latter offering slightly better speed. This underscores the balance between embedding dimensions, quantization bits, retrieval performance, and speed, highlighting the potential for significant speed-ups with only modest reductions in accuracy.

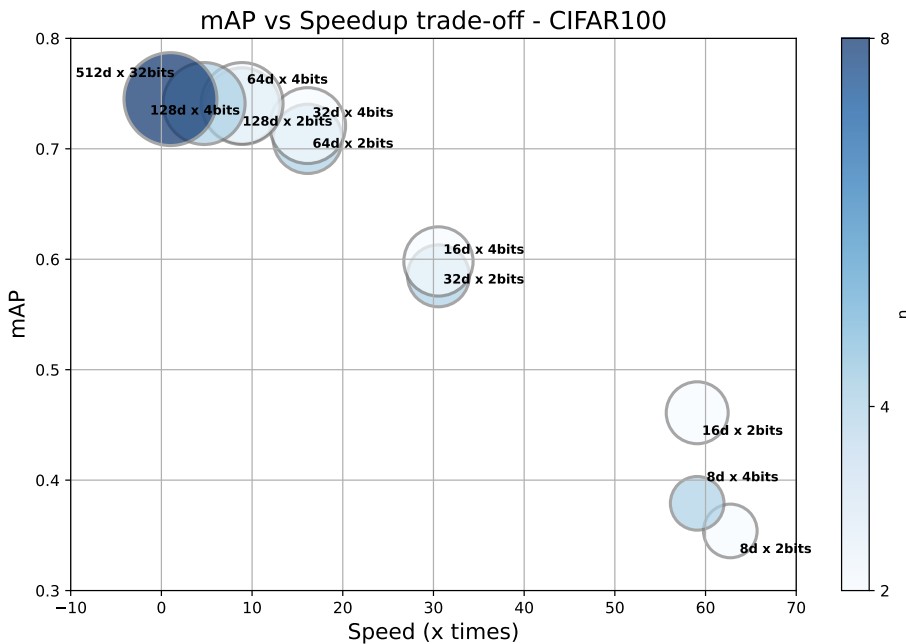

Figure 6: **The effect of embedding dimensions and quantization bits** on CIFAR100 on CLIP visual backbone. The darkest bubble denotes the full-precision Euclidean embedding. We find it is best to use more dimensions with strong compression than vice versa. With binary hyperbolic embeddings, we can obtain > 4.7 times faster retrieval speeds at roughly the same performance. Naturally, we can compress much further, at the cost of retrieval performance.

### E.2 EFFECT OF HIERARCHICAL KNOWLEDGE

To measure this potential we follow Long et al. (2020); Ghadimi Atigh et al. (2021) in using the *Sibling mAP* (SmAP) performance metric. Building upon the mAP metric, SmAP takes into account the proximity in the class hierarchy for retrieved items. Specifically, when an item retrieved is just one hop away (i.e., same parent class) from the ground truth it is considered a true positive.

Table 8: **The effect of using hierarchical knowledge and different manifolds** on CIFAR100 on CLIP visual backbone. At full-precision, Spherical and hyperbolic embeddings outperform Euclidean embeddings, at the cost of a large number of bits and/or slow distance calculations. By binarizing we can maintain the performance benefits of hierarchy and hyperbolic embeddings but at a highly compressed bit length.
$\star$ denotes that the method has been modified to be using hyperbolic distance metric.

|  | Hierarchical | Manifold | Bit length | Binary | mAP | SmAP |
|---|---|---|---|---|---|---|
| Radford et al. Radford et al. (2021) | ✗ | $\mathbb{R}$ | 16,384 | ✗ | 0.721 | 0.835 |
| Kasarla et al. Kasarla et al. (2022) | ✗ | $\mathbb{S}$ | 3,168 | ✗ | **0.746** | 0.860 |
| Kasarla et al. Kasarla et al. (2022)$\star$ | ✗ | $\mathbb{D}$ | 3,168 | ✗ | 0.744 | 0.863 |
| Barz & Denzler Barz & Denzler (2020) | ✓ | $\mathbb{S}$ | 3,200 | ✗ | 0.719 | 0.843 |
| Long et al. Long et al. (2020) | ✓ | $\mathbb{D}$ | 1,600 | ✗ | **0.746** | **0.868** |
| Binary (ours) | ✓ | $\mathbb{D}$ | 512 | ✓ | 0.745 | 0.866 |

In Table 8 and 9 we perform a comparison to gain insight into the effect of using hierarchy and hyperbolic embeddings. We show that prior full-precision approaches perform comparable in terms of mAP when using a hyperbolic or Spherical manifold, and that adding hierarchy is especially beneficial to the SmAP performance. However, despite being more compressed than full-precision Euclidean manifolds these approaches still require a large number of bits and/or slow hyperbolic

distance calculations. By binarizing, we are able to compress hierarchical hyperbolic embeddings to a small bit length whilst maintaining good performance on both standard and hierarchical metrics.

Table 9: **The effect of using hierarchical knowledge and different manifolds** on ImageNet1K on CLIP visual backbone. At full-precision Spherical and hyperbolic embeddings outperform Euclidean embeddings, at the cost of a large number of bits and/or slow distance calculations. By binarizing we can maintain the performance benefits of hierarchy and hyperbolic embeddings but at a highly compressed bit length.
$\star$ denotes that the method has been modified to be using hyperbolic distance metric.

|  | Hierarchical | Manifold | Bit length | Binary | mAP | SmAP |
|---|---|---|---|---|---|---|
| Radford et al. Radford et al. (2021) | ✗ | $\mathbb{R}$ | 16,384 | ✗ | 0.593 | 0.787 |
| Kasarla et al. Kasarla et al. (2022) | ✗ | $\mathbb{S}$ | 3,168 | ✗ | 0.607 | 0.802 |
| Kasarla et al. Kasarla et al. (2022)$\star$ | ✗ | $\mathbb{D}$ | 3,168 | ✗ | 0.607 | 0.807 |
| Barz & Denzler Barz & Denzler (2020) | ✓ | $\mathbb{S}$ | 3,200 | ✗ | 0.599 | 0.811 |
| Long et al. Long et al. (2020) | ✓ | $\mathbb{D}$ | 1,600 | ✗ | **0.610** | **0.815** |
| Binary (ours) | ✓ | $\mathbb{D}$ | 512 | ✓ | 0.608 | 0.812 |

### E.3 EFFECT OF CURVATURE AND RADIUS

In Table 10, we explore the effect of curvature on the ImageNet1K dataset. The parameter $r^2$ represents the squared radius used for constructing class prototypes on the Poincaré disk, while $c$ denotes the curvature used when optimizing the function to map data samples to class prototypes. Low curvature values indicate almost flat, Euclidean-like space, whereas high curvature values can cause numerical instability. Therefore, a medium curvature value is preferred for both class prototype embedding and sample embedding.

Table 10: **The effect of curvature** on ImageNet1K on CLIP visual backbone. Parameter $r^2$ is the squared radius used when constructing the class prototypes $\boldsymbol{P}$ on poincaré disk, whereas $c$ is the curvature used when optimizing $f(\cdot)$ to map data samples to class prototypes. Low curvature indicates almost uncurved, euclidean-like space, whereas high curvature causes numerical instability. Thus, a medium value of curvature during both class prototype embedding and sample embedding will be preferred.

|  | $r^2 = 10^3$ | $r^2 = 10^2$ | $r^2 = 10$ | $r^2 = 1$ | $r^2 = 0.1$ |
|---|---|---|---|---|---|
| $c = 10^{-3}$ | 0.493 | 0.521 | 0.587 | 0.596 | 0.588 |
| $c = 10^{-2}$ | - | 0.586 | 0.599 | **0.608** | 0.594 |
| $c = 0.1$ | - | - | 0.601 | 0.607 | 0.605 |
| $c = 1$ | - | - | - | 0.549 | 0.550 |
| $c = 10$ | - | - | - | - | 0.372 |

The table lists the accuracy results for different combinations of $r^2$ and $c$. For $r^2 = 10^3$, the accuracy improves from 0.493 with $c = 10^{-3}$ to 0.521 with $r^2 = 10^2$ and further to 0.587 with $r^2 = 10$. The best performance, with an accuracy of 0.608, is achieved with $r^2 = 1$ and $c = 10^{-2}$. As $r^2$ decreases further to 0.1, the accuracy decreases slightly. High curvature values ($c = 1$ and $c = 10$) result in significantly lower accuracy, highlighting the numerical instability.

In summary, medium curvature values during both class prototype embedding and sample embedding yield better performance, while high curvature causes numerical instability and lower accuracy.

## F    MORE RELATED WORK

**Hyperbolic representation**   differs from the Euclidean representation commonly used in deep learning. Early success was obtained by embedding the nodes of hierarchies as hyperbolic vectors, outperforming Euclidean embeddings. Nickel & Kiela (2017) introduce Poincaré Embeddings, where hierarchical nodes are positioned by pulling and pushing nodes based on parent-child relations. Ganea et al. (2018b) extend this idea through hyperbolic entailment cones, where child nodes

should strictly fall under the cone spanned by parent nodes. Other hyperbolic embeddings Sala et al. (2018); Balazevic et al. (2019); Tseng et al. (2023); Yu et al. (2024) further explore these ideas for incomplete information, graphs, and generalibility, respectively. To make the step towards deep learning in hyperbolic space, Ganea et al. (2018a) and Shimizu et al. (2021) introduce hyperbolic linear, recurrent, convolutional, and self-attention layers in the most commonly used model of hyperbolic space: the Poincaré ball model. These works have served as a foundation for hyperbolic deep learning on graphs Pan & Wang (2021), dimensionality reduction Chami et al. (2021), complex networks Muscoloni et al. (2017), social media Sawhney et al. (2021), etc. For more details on hyperbolic layers, we refer to the survey of Peng et al. (2021).

Hyperbolic learning has also been broadly investigated in the image and video domain Mettes et al. (2023). Hyperbolic geometry has been shown to aid in a variety of tasks, including image segmentation Atigh et al. (2022), object detection Lang et al. (2022), action recognition Long et al. (2020); Peng et al. (2020), image-text representation learning Desai et al. (2023), representing 3D point clouds Montanaro et al. (2022) and LiDAR pose regression Wang et al. (2023). These works have in common that they leverage hierarchical knowledge to maximize the benefits from the hyperbolic space, embedding related concepts closer together, thereby allowing for more compact and powerful representations.

