# OpenReview forum: "Binary Hyperbolic Embeddings"
_ICLR.cc/2025/Conference — Submitted to ICLR 2025_

### Official Review · Reviewer_sdRm · 2024-11-01

**Soundness:** 2
**Presentation:** 2
**Contribution:** 3
**Rating:** 6
**Confidence:** 3

**Summary:**

This paper proposes a method, Binary Hyperbolic Embeddings, that combines the advantages of Binary Euclidean and Hyperbolic embeddings, offering both speed and low dimensionality.

**Strengths:**

The paper is highly original and novel, with well-substantiated arguments and no overclaims.

**Weaknesses:**

1. Some tables are not expressed appropriately; for instance, using "s" to indicate speed in Table 3.
2. Some figures are not expressed appropriately; for example, in Figure 1, the image comparing Binary Hyperbolic and Hyperbolic embeddings does not clearly convey the concept of "Binary."

**Questions:**

1. What is the cost (in terms of time and space complexity) when transitioning from Euclidean to Binary Hyperbolic?
2. In which other scenarios does this method still perform less effectively?

---

> ### Author Response · Authors · 2024-11-28
> **Response to Reviewer sdRm**
>
> ### **Response to Reviewer sdRm**
>
> We especially thank the reviewer for appreciating that our work is **"highly original and novel"**. We thank you for your detailed feedback and constructive suggestions. Below, we address your comments and questions:
>
> ---
>
> **Weaknesses**
>
> **[W1]: *Tables and speed representation***
>
> **[Update]:** We have revised Table 2 by replacing "speed" with "time" for clarity. Additionally, we updated Table 1 to represent speed in relative terms (e.g., as multiples of the baseline, denoted by $\times$).
>
> **[W2]: *Figures and visualization of binarization***
>
> **[Update]:** We have updated Figure 1 to include a discrete grid overlay on the Poincaré disk, illustrating the concept of binarization. Similarly, in Figure 2, we now use a discrete grid representation to depict the binarized Poincaré ball for hyperbolic space. Please note that these visualizations are designed for illustrative purposes, as real binarization occurs in high-dimensional space and involves binarizing along specific latent dimensions, not directly on the axis or radius.
>
> ---
>
> **Questions**
>
> **[Q1]: *Cost of transitioning from Euclidean to Binary Hyperbolic***
>
> **[A1]:** The cost of transitioning involves both **time and space complexity**. We analyzed the time complexity of operations in hyperbolic space, including the Möbius addition and exponential/logarithmic maps, in Appendix A.6. Additionally, spatial complexity increases due to the nature of hyperbolic computations. Specifically, as discussed in [r1], even with optimization techniques (e.g., pre-computation), Möbius addition requires **approximately 2× the memory** of its Euclidean counterpart. These trade-offs are further elaborated in the updated appendix.
>
> **Reference:**
>
> [r1] Atigh, Mina Ghadimi, et al. "Hyperbolic image segmentation." Proceedings of the IEEE/CVF Conference on Computer Vision and Pattern Recognition. 2022.
>
> **[Q2]: *Scenarios where the method underperforms***
> **[A2]:** Binary hyperbolic embeddings, like hyperbolic embeddings in general, are less effective when the underlying data does not exhibit a clear hierarchical structure. In such cases, the benefits of hyperbolic geometry in capturing hierarchical relationships diminish.
>
> ---
>
> We appreciate your valuable feedback, which has helped improve the clarity and rigor of our submission. We hope these updates address your concerns and provide additional insight into the strengths and limitations of our method.

---

> > ### Comment · Reviewer_sdRm · 2024-12-02
> >
> > Thank you for addressing my questions. I decide to keep the score.

---

### Official Review · Reviewer_p8o6 · 2024-11-01

**Soundness:** 3
**Presentation:** 3
**Contribution:** 3
**Rating:** 6
**Confidence:** 4

**Summary:**

This paper introduces Binary Hyperbolic Embeddings, an approach to vector similarity search that take the advantages of binarised hyperbolic space. The objective is to combine the fast retrieval capabilities of binary operations with the compact and hierarchical properties of hyperbolic embeddings. While hyperbolic space is effective to capture hierarchical relationships, its practical applications show low computational efficiency of metric calculations. This study addresses these challenges by transforming hyperbolic similarity metrics into binary Hamming distance operations, enabling faster and more efficient retrieval.

**Strengths:**

1. A key contribution in this paper is the proof of retrieval equivalence. The authors show that search results obtained using hyperbolic distance remain identical when replaced with Hamming distance, given certain conditions. This allows the method to maintain the quality of retrieval outcomes while significantly reducing computational overhead.

2. Another key contribution is the integration of their approach with FAISS. By combining their binary hyperbolic method within FAISS, the authors present improvements in both memory usage and processing speed, without compromising the precision achieved with full-precision embeddings.

3. The paper also highlights the robustness of binary hyperbolic embeddings in preserving retrieval quality across a wide range of datasets. This characteristic is particularly beneficial for tasks that involve hierarchical structures, such as image or video retrieval. They evaluate the performance across several datasets, including CIFAR100, ImageNet1K, and Moments in Time.

4. Additionally, the study explores the trade-offs between memory efficiency, retrieval speed, and accuracy. Their experimental results show that binary hyperbolic embeddings not only meet but often surpass the performance of other state-of-the-art methods.

Overall, Binary Hyperbolic Embeddings present a practical and powerful solution for similarity search, especially in large-scale scenarios where fast processing and memory optimisation are essential. This approach successfully bridges the gap between the theoretical strengths of hyperbolic space and the practical demands for efficient, scalable retrieval systems.

**Weaknesses:**

For experiment:
1.	In the captain of Table 1, “Our binary hyperbolic embeddings at 512 bits can maintain this performance while being much faster to evaluate, thereby getting the best of both worlds.”  I cannot find the time cost comparison of different methods regarding these three datasets.
2.	In line 346-347, “On ImageNet1K, we obtain a mAP@50 of 63.44% compared to 63.24% and 63.14% for the Euclidean and hyperspherical baselines.”  I am confused on what this statement is trying to show. Statistically, the mAP of 0.6344 (std 0.0011) is not significantly better than 0.6314 (std 0.019) or 0.6324 (std 0.0037). You might present more description for this statement, or use other results to favour your hypothesis.

For theory:
Since the binary hyperbolic embeddings is an approximate method, it is worthy to discuss its relationship with the sketching methods such as LSH methods.

**Questions:**

Can you discuss the relationship of binary hyperbolic embeddings with sketching methods such as LSH and random projection?

Minor comments:
1.	SmAP is not defined in the main manuscript, but in Appendix.
2.	Figure 3 is not discussed in the main context.
3.	Line 368-369, no Figure 4.3 found in manuscript.
4.	Line 369-370, “The trade-off between embedding dimensions and quantization levels on ImageNet1K.” is not a full sentence.

---

> ### Author Response · Authors · 2024-11-26
> **Response to Reviewer p8o6**
>
> ## Response to Reviewer p8o6
>
> We appreciate the reviewer’s recognition on several our contributions, particularly the **proof of retrieval equivalence**. We are pleased that the **integration** **binary hyperbolic embeddings with FAISS** and the resulting improvements in **memory usage** and **processing speed** without compromising retrieval quality were highlighted. Your acknowledgment of the method’s **robustness** across datasets is encouraging. Thank you for emphasizing our exploration of the **memory-speed-accuracy trade-offs** and the practicality of our approach for **large-scale similarity search**.
>
> ---
>
> ### **Response to Experimental Comments**
>
> 1. **Time Cost Comparison (Table 1)**
>
>     We summarize the speed up in the following table.
>     **[Update]:** we also update Table 1 and corresponding text accordingly.
>
> | Manifold         | Embedding size | CIFAR100 Speed | ImageNet1K Speed | Moments-in-Time Speed |
> |------------------|----------------|----------------|------------------|-----------------------|
> | $\mathbb{R}^D$   | 16384 bits     | 1.00x          | 1.00x            | 1.00x                 |
> | $\mathbb{S}^d$   | 8192 bits      | 0.99x          | 1.01x            | 1.00x                 |
> | $\mathbb{D}^d$   | 8192 bits      | 0.22x          | 0.18x            | 0.21x                 |
> | $\mathbb{R}^B$   | 1024 bits      | 2.21x          | 2.28x            | 2.29x                 |
> | $\mathbb{S}^b$   | 512 bits       | 4.19x          | **8.25x**            | 4.47x                 |
> | $\mathbb{D}^b$ (ours) | 512 bits  | **4.20x**      | 8.24x        | **4.50x**             |
>
>
> 2. **irrelavent mAP@50 description on ImageNet1K**
>     We removed the narration as it is less relavent to the focus of binary hyperbolic embeddings.
>
> ---
>
> ### **Reponse to Theoretical Comment & Question**
>
> 1. **Relationship with Sketching Methods (e.g., LSH)**
>
> **[Q]: Can you discuss the relationship of binary hyperbolic embeddings with sketching methods such as LSH and random projection?**
> Both sketching methods and binary hyperbolic embeddings aim to efficiently process and analyze large-scale datasets, yet they differ in their principles, geometry, and use cases:
>
> ---
>
> - **1. Geometry**
>     - **Sketching Methods**: Primarily operate in Euclidean geometry, leveraging probabilistic hashing, counting, and random projections to approximate similarity metrics.
>     - **Binary Hyperbolic Embeddings**: Rooted in hyperbolic geometry, which inherently supports low-dimensional representations and efficiently captures hierarchical structures.
>
> - **2. Underlying Principle**
>     - **Sketching Methods**: Approximation is achieved through techniques like probabilistic hashing (e.g., LSH), frequency sketches, or random projections.
>     - **Binary Hyperbolic Embeddings**: Leverage the compactness of hyperbolic space to represent data in a way that preserves its hierarchical relationships, further refined through binary quantization.
>
> - **3. Approximation Error**
>     - **Sketching Methods**: Approximation errors often arise due to hash collisions, particularly for complex or high-dimensional data distributions.
>     - **Binary Hyperbolic Embeddings**: Errors are introduced via binary quantization.
>
> -  **4. Specialization**
>     - **Sketching Methods**: Well-suited for approximate similarity search in flat, Euclidean spaces, particularly for datasets without significant structural hierarchy.
>     - **Binary Hyperbolic Embeddings**: Inherit the advantages of hyperbolic space, including the ability to preserve hierarchical or graph-like structures, making them ideal for tasks involving tree-like or nested relationships.
>
> ---
> ### **Response to Minor Comments**
>
> 1. **SmAP Definition**
>    Thank you for pointing out the omission. We have now defined SmAP in the main manuscript for clarity.
>
> 2. **Figure 3 Discussion**
>    We appreciate your observation. We have added a discussion of **Figure 3** in the main context to connect it to our analysis.
>
> 3. **Reference to "Figure 4.3"**
>    This was an oversight, and we have corrected the reference to point to the appropriate figure in the manuscript.
>
> 4. **Incomplete Sentence (Line 369-370)**
>    Thank you for identifying this issue. We have revised the sentence to read:
>    "Figure 4 shows the trade-off between embedding dimensions and quantization levels on ImageNet1K"
>
> ---
>
> ### **[Updates]:**
> We have incorporated your feedback into the revised manuscript in Section 4, addressing the experimental weaknesses, and fixing the minor issues identified. Thank you for helping us improve the clarity and rigor of our work.

---

> > ### Comment · Reviewer_p8o6 · 2024-11-27
> >
> > Thank you for addressing my questions. I decide to keep the score.

---

### Official Review · Reviewer_Fcqp · 2024-11-03

**Soundness:** 1
**Presentation:** 2
**Contribution:** 1
**Rating:** 1
**Confidence:** 5

**Summary:**

The paper presents a solution for using quantized hyperbolic embeddings for vector search. To the best of my understanding the authors do not seem to be proposing a new model for computing the hyperbolic embeddings, and follow Long et al. (2020) for this. The innovation resides in the binarization to overcome the inefficiencies of the hyperbolic distance computations. With the proposed approach, memory is spared and speed is gained while maintaining accuracy.

**Strengths:**

- The authors show that quantizing hyperbolic embeddings leads to almost no decrease in accuracy in the supervised vector search setting. Even going as low as using 2 bits per value seems to provide good results
- The authors provide a sound comparison showing that using hyperbolic embeddings yields better accuracy and compression than using linear or spherical maps when dealing with labeled data with a hierarchical structure.

**Weaknesses:**

First of all, I believe that the overall presentation is slightly misleading. By reading the abstract, introduction, and most of the paper, it seems that this is a general method that could replace any of the standard embedding methods in vector search. However, it is clear in Appendix A that the authors are only dealing with the supervised setting. This should be clearly stated. Moreover, the use of vector search in research and industry for GenAI is moving away from the supervised setting. This does not imply that this setting is not interesting in itself, but it serves a narrower scope and this should be clearly stated.

More importantly, the whole work hinges on the statement that "calculating the distance between embeddings [in the hyperbolic space] involves slow vector operations." Although true strictly speaking when looking at Equation (2), I argue from two different perspectives that this is not a constraint in practice.

- Perspective 1: If one can replace one distance with another that is easier to compute and that preserves ordering, one can perform vector search. Here, only the total order induced by the distance to the query matters. The authors use the well known relationship in Equation (14) of Appendix B, found in (Ganea et al., 2018a) for example. This lead to Equation (15):

$$||q - x_1||^2 / (1 - ||x_1||^2) < ||q - x_2||^2 / (1 - ||x_2||^2)$$

where I have omitted the term $1 - ||q||^2$ on both sides as it does not change the total order. The authors should explain why the dissimilarity $||q - x||^2 / (1 - ||x||^2)$ is such a computational burden. In the worst case scenario that we do not know $||x||$ (more on this next), it requires twice the number of multiplications than the squared Euclidean distance. Given modern arithmetic speed in CPUs and GPUs, once the data has been fetched  from memory this 2x increase seems trivial. If this is not the case, the authors should clearly explain why it is not the case.

- Perspective 2: The authors seem to be using class membership as the ground truth (please correct me if I'm wrong). The angle between two vectors in hyperbolic space is sufficient to determine the class they belong to. This observation was also done by Long et al (2020), where the $1 - \cos(q,x)$ is used as the dissimilarity measure. This means that we could pre-normalize all vectors in the database and simply use $1 - <q,x>$ as the dissimilarity measure. Using fused-multiply-add instructions, this calculation is extremely fast.

Notice that these two perspectives do not require any type of conditions or assumptions on the embedding vectors, contrarily to Proposition 3.1.

The main condition in Proposition 3.2 states that $<x,y>$ should be proportional to $<g(x),g(y)>$. Then, the derivation in equation (26) of the appendix does not lead to proportionality of the squared distance, but to inequalities for it since the different terms will have different proportionality constants. For example,
- $<x,x> \propto <g(x),g(x)>$ with scalar constant $a$
- $<y,y> \propto <g(y),g(y)>$ with scalar constant $b$
- $<x,y> \propto <g(x),g(y)>$ with scalar constant $c$

This will lead to prove either that $d_R^2(x,y) \leq d_H(x,y)$ or that $d_R^2(x,y) \geq d_H(x,y)$ depending on $a, b, c$. Then the Proposition stops being useful, unless the constants are tight, which requires a much more detailed analysis.

However, the use of the symbol $\propto$ in Equation (9), may indicates that the authors are using it to indicate proximity (commonly noted $\approx$) and not proportionality (commonly noted $\propto$). If it is proximity , then Proposition 3.2 does not seem to be very useful. It boils down to stating that g is such that if $x \approx g(x)$, then $d_R^2(x,y) \approx d_H(x,y)$, which is trivial unless you characterize mathematically the nature of the proximity (as was done in locality-sensitive hashing (LSH), for example).

In any case, the word equivalence is too strong for this type of result. That is why the seminal LSH is considered an approximation.

Regarding the binary quantization, I do not understand the rationale. Coming back to my previous paragraph about notation ($\approx$ versus $\propto$), maybe the authors mean something different than what they actually wrote in Equation (9). Because of the quantization step in Equation (7), $<x,y>$ is not proportional to $<x_{int}, y_{int}>$. Moreover, it seems that the authors are reinventing base 2 integer arithmetics. Any modern CPU/GPU can perform $<x_{int}, y_{int}>$ with extreme efficiency at the hardware level using intrinsic SIMD instructions. For example, for vectors with 512 or 256 bits, this computation can be done with a single instruction on modern CPUs. If using less than 8-bit quantization, this will require a few more instructions but the point still stands). However, the authors opt to bypass it with the software implementation in Equation (10), which clearly needs many CPU instructions, even for 256-bits or 512-bit vectors.

Minor editorial comments:
- In Section 2, there are quite a few references that do not use the proper citet/citep format. This complicates reading these paragraphs. This occurs in other parts of the paper as well.
- In line 368, the reference to "Figure 4.4" is wrong. It should be "Figure 4"
- There are figures that are never been referenced in the text, Figure 2 for example. Please make sure that all figures and tables are properly referenced in the main body.
- Footnote 1 on page 4 refers to Appendix D and it should refer to Appendix C.
- There seem to be missing terms in Equation (9). Maybe the authors missed writing "+ ... +" before the last term.
- Please provide two contrasting colors and different line styles for the curves in Figure 3.

**Questions:**

- Is there a way to apply the proposed approach in the unsupervised setting?
- Please explain and clarify whether the inefficiencies of the hyperbolic distance computations are truly a bottleneck for the application addressed in this paper.
- Please clarify the mathematical notation ($\approx$ versus $\propto$), and be more precise in the language of the proposition (find a better term than equivalence)
- Please evaluate whether Proposition 3.1 has a simpler alternative in line of the arguments I provided.
- Please evaluate whether binarization (Proposition 3.2 and Section 3.3) is even needed, as working with integers seems to be more efficient than decomposing their arithmetic.
- Please clarify how the training is done regarding the data (separation into training/evaluation/test sets, etc.)
- The common operation in scalar quantization is to perform rounding (e.g., floor(x + 0.5)) instead of truncation as it yields a lower quantization error. Why are the authors using truncation in Equation (7)?

---

> ### Author Response · Authors · 2024-11-25
> **Response to Reviewer Fcqp(1/3)**
>
> ## Response to Reviewer Fcqp(1/3)
>
> We thank the reviewer recognition with our binarization, ``memory is spared and speed is gained while maintaining accuracy.``
>
> We give responses as follows:
>
> $$$$
>
> **[W1]** *Vector search in research and industry for GenAI is moving away from the supervised setting. While this does not imply that the supervised setting is uninteresting, it serves a narrower scope, and this should be clearly stated.*
> **[Q1]** *Is there a way to apply the proposed approach in the unsupervised setting?*
>
> ---
>
> ### **Response to W1, Q1**
> Thank you for the observation. While our work focuses primarily on the supervised setting, we do consider an unsupervised configuration. Specifically, in **Table 5 (upper half)**, we present results under an unsupervised setup. This approach aligns closely with [r1], where self-supervised contrastive learning is applied to the backbone, combined with a hyperbolic head.
>
> **Reference**
> [r1] Wei, Rukai, et al. "Exploring Hierarchical Information in Hyperbolic Space for Self-Supervised Image Hashing." *IEEE Transactions on Image Processing* (2024).
>
> ---
>
> ### **[Updates]**
> We notice that in the era of GenAI, **supervised** fine-tuning remains a critical component. To address this and provide additional clarity:
> - We have revised the **Introduction** and **Experimental Setup** sections to explicitly state our primary focus on the supervised setting.
> - We have updated the title to:  *"Supervised Binary Hyperbolic Embedding"* to better reflect the scope of our work.
>
> We also plan to expand our work with additional unsupervised experiments and will provide updates in follow-up comments once these results are ready.
>
> $$$$
>
> **[W2, Q2]** *Calculating the distance between embeddings [in hyperbolic space] involves slow vector operations.*
>
> ---
>
> **[W2, Prospective 1]:**
> The distance computation can be expressed as:
> $$ \frac{\|q - x_1\|^2}{1 - \|x_1\|^2} < \frac{\|q - x_2\|^2}{1 - \|x_2\|^2} $$
> In the worst-case scenario where $\|x\|$ is unknown, this calculation requires twice as many multiplications as the squared Euclidean distance. However, given the modern arithmetic capabilities of CPUs and GPUs, this 2x increase is negligible once the data has been fetched from memory.
>
> ---
>
> ### **Response to W2, Prospective 1**
> We appreciate the reviewer for highlighting this memory perspective, as it aligns with our key strength: **low memory requirement**. Specifically, our method involves exhaustive search, meaning all vectors are already loaded into memory. Therefore, this 2x computational overhead occurs entirely within memory operations.
>
> ### **Key Strength: Low Memory Requirement**
> As shown in **Table 2**, our method's **extremely low memory footprint** enables significant scalability advantages. This makes the acceleration a natural by-product of our low-bit representation. Importantly, **even without considering acceleration**, the low-memory benefit remains compelling.
>
> For instance, when scaling to **~1 billion vectors**, our binary representation can hold all vectors within **64GB of memory**. In contrast, approximate nearest neighbor (ANN) methods cannot perform exhaustive search with such low memory consumption.
>
> ---
>
> ### **[Updates]**
> - We have expanded our discussion in the **Related Work** section, emphasizing the relationship between our method and ANN approaches, including works such as **DiskANN** [r2], which requires external storage.
> - We analyzed the complexity of distance computation in terms of the number of operations in **Appendix A.6**.
>
> ---
>
> **Reference**
> [r2] Jayaram Subramanya, Suhas, et al. "DiskANN: Fast accurate billion-point nearest neighbor search on a single node." *Advances in Neural Information Processing Systems 32* (2019).

---

> > ### Comment · Reviewer_Fcqp · 2024-11-27
> > **Response**
> >
> > Although Table 5 presents unsupervised experiments, they are not convincing. There are methods implemented in FAISS (OPQ, for example) that yield better accuracy than PQ. The authors should pick strong baselines. Regarding memory savings, Table 5 shows that in the unsupervised scenario the proposed method performs on par with PQ using the same footprint. I see no improvement over the baseline.
> >
> > Regarding the 2x overhead of using $||q - x_1||^2 / (1 - ||x_1||^2) < ||q - x_2||^2 / (1 - ||x_2||^2)$, it is clearly possible to store the norm of each vector in memory at the cost of 2 bytes per vector (in FP16 format) and have a computational cost equivalent to that of an Euclidean distance. The authors claim that these computations are slow, when they are not.
> >
> > Regarding the claims about scalability to 1 billion vectors, the authors cannot hypothesize about it and claim it as a win without actually running experiments. There are may standard datasets for this in https://big-ann-benchmarks.com/neurips23.html.

---

> ### Author Response · Authors · 2024-11-25
> **Response to Reviewer Fcqp(2/3)**
>
> ## Response to Reviewer Fcqp(2/3)
>
> **[W2, Prospective 2:]** *The angle between two vectors in hyperbolic space is sufficient to determine the class they belong to. This observation was also noted by Long et al. (2020)... this $1 - \cos(q, x)$ calculation is extremely fast.*
>
> ---
>
> ### **Response to W2, Prospective 2**
> We acknowledge that $1 - \cos(q, x)$ can serve as a fast approximation to the hyperbolic distance $d_\mathbb{D}(q, x)$. However, this approach has two significant drawbacks:
>
> 1. **Incompatibility with Binary Embeddings**
>    Cosine similarity is not compatible with binary embeddings, which form the foundation of our contributions in terms of both **acceleration** and **memory reduction**. This is evident in the following comparison using Faiss:
>
> | Embedding                   | mAP@50 ↑          | Index Size ↓ | Retrieval Time (s) ↓ |
> |-----------------------------|-------------------|--------------|-----------------------|
> | Euclidean-256D             | 0.5847 ± 0.0040  | 51MB         | 1.11 ± 0.04          |
> | Hyperbolic (cosine)-256D    | 0.6344 ± 0.0011  | 51MB         | 1.11 ± 0.06          |
> | BinaryHyperbolic-256bit    | 0.6320 ± 0.0014  | **1.5MB**    | **0.28 ± 0.03**      |
> | BinaryEuclidean-512bit     | 0.5849 ± 0.0024  | 3MB          | 1.03 ± 0.04          |
> | **BinaryHyperbolic-512bit** | **0.6358 ± 0.0006** | **3MB**      | **1.04 ± 0.03**      |
>
> This table demonstrates that even cosine distance based hyperbolic is substantially slower than binary counterparts.
>
> **[W2, Assumptions]** *two perspectives do not require any type of conditions or assumptions on the embedding vectors*
>
> 2. **Lack of Ranking Preservation**
>    Cosine similarity, without additional conditions, **does not preserve ranking in hyperbolic space**. Specifically, it cannot distinguish between vectors with different radii (e.g., $||q_1|| = r_1$ vs. $||q_2|| = r_2$). On the Poincaré disk, the radius conveys crucial information about embeddings, and losing this information can lead to **ranking inconsistencies**, which is critical for retrieval tasks where accurate ordering of results is essential.
>
> $$$$
>
> **[W3.1]** *...the derivation in equation (26) of the appendix does not lead to proportionality of the squared distance.*
>
> ---
>
> ### **Response to W3.1**
> We believe this concern may stem from a misunderstanding. To clarify, **proportionality is a pre-condition of this proposition**, rather than an intermediate result or conclusion. If this pre-condition holds, then equation (26) (now equation (44)) follows accordingly.
>
> ---
>
> ### **[Updates]:**
> To address any potential ambiguity, we have revised the notation in equation (26) (now equation (44)) in the appendix for improved clarity.
>
> $$$$
>
> **[W3.2]** *Different proportional factors $a, b, c$ may lead to different conclusions $d_R^2(x, y) \leq d_H(x, y)$ or $d\_R^2(x, y) \geq d_H(x, y)$.*
>
> ---
> ### **Response to W3.2**
> We believe this point may not be directly relevant to our paper. In our approach, there are no varying proportional factors $a, b, c$ for the same embedding. The quantization scale is fixed on a Poincaré disk with radius $r$. Given this fixed quantization scale, only a single proportional factor exists.
>
> We would appreciate it if the reviewer could provide a more concrete example to clarify this concern further.

---

> > ### Comment · Reviewer_Fcqp · 2024-11-27
> > **Proportionality in Proposiiton 3.2**
> >
> > The claims about proportionality continue to not make sense to me. The quantizer eliminates any hope of proportionality. How are the authors quantizing the data (some numbers are rounded up, and some number are rounded down) and yet maintaining proportionality?

---

> ### Author Response · Authors · 2024-11-25
> **Response to Reviewer Fcqp(3/3)**
>
> ## Response to Reviewer Fcqp(3/3)
>
> **[W4, Q3]** *However, the use of the symbol $\propto$ in Equation (9) may indicate that the authors are using it to denote proximity (commonly noted $\approx$) and not proportionality (commonly noted $\propto$).*
>
> ---
>
> ### **Response to W4, Q3**
> We confirm that we indeed mean **proportional to** ($\propto$), not proximity ($\approx$). The confusion might stem from overlooking our claim regarding the **block-wise application of Proposition 3.2**. To clarify, here is an illustrative example:
>
> ---
>
> ### **Example**
> Given:
> $$
> \mathbf{x}= \begin{pmatrix} 0.50 \\\\ 0.75 \end{pmatrix}, \quad \mathbf{y}= \begin{pmatrix} 0.25 \\\\ 0.50 \end{pmatrix}
> $$
>
> Quantizing each dimension with 2 bits on $(0, 1)$, the quantization scale is $s = 0.25$. This results in:
>
> $$
> \mathbf{x}\_{int}= \begin{pmatrix} 2 \\\\ 3 \end{pmatrix}, \mathbf{y}_{int}= \begin{pmatrix} 1 \\\\ 2 \end{pmatrix}
> $$
>
> We can decompose $\mathbf{x}$ and $\mathbf{y}$ into block-wise components:
> $$
> \mathbf{x}= \begin{pmatrix} 0.50 \\\\ 0.75 \end{pmatrix} = \mathbf{x}_1 + \mathbf{x}_2 = \begin{pmatrix} 0.50 \\\\ 0.50 \end{pmatrix} + \begin{pmatrix} 0.00 \\\\ 0.25 \end{pmatrix}
> $$
> $$
> \mathbf{y}= \begin{pmatrix} 0.25 \\\\ 0.50 \end{pmatrix} = \mathbf{y}_1 + \mathbf{y}_2 = \begin{pmatrix} 0.00 \\\\ 0.50 \end{pmatrix} + \begin{pmatrix} 0.25 \\\\ 0.00 \end{pmatrix}
> $$
>
> Here, $\mathbf{x}\_1$ corresponds to the more significant bit, and $\mathbf{x}\_2$ corresponds to the less significant bit. Thus, their block-wise binary representations are:
>
> $$\mathbf{x}_{int} = \begin{pmatrix} 2 \\\\ 3 \end{pmatrix} = 2^1 \cdot \mathbf{x}_1^b + 2^0 \cdot \mathbf{x}_2^b = 2^1 \cdot \begin{pmatrix} 1 \\\\ 1 \end{pmatrix} + 2^0 \cdot \begin{pmatrix} 0 \\\\ 1 \end{pmatrix}$$
>
> $$ \mathbf{y}_{int} = \begin{pmatrix} 1 \\\\ 2 \end{pmatrix} = 2^1 \cdot \mathbf{y}_1^b + 2^0 \cdot \mathbf{y}_2^b = 2^1 \cdot \begin{pmatrix} 0 \\\\ 1 \end{pmatrix} + 2^0 \cdot \begin{pmatrix} 1 \\\\ 0 \end{pmatrix}
> $$
>
> This gives the following **block-wise proportional relationship**:
>
> $$
> \begin{align}
> \langle \mathbf{x}, \mathbf{y} \rangle &\propto \langle \mathbf{x}\_{int}, \mathbf{y}_{int} \rangle \\\\
> \langle \mathbf{x}_1, \mathbf{y}_1 \rangle &\propto \langle \mathbf{x}_1^b, \mathbf{y}_1^b \rangle \\\\
> \langle \mathbf{x}_2, \mathbf{y}_2 \rangle &\propto \langle \mathbf{x}_2^b, \mathbf{y}_2^b \rangle \\\\
> &\vdots \\\\
> \langle \mathbf{x}_n, \mathbf{y}_n \rangle &\propto \langle \mathbf{x}_n^b, \mathbf{y}_n^b \rangle
> \end{align}
> $$
>
> Concatenating binary components yields:
> $$
> \mathbf{x}^b = \begin{pmatrix} \mathbf{x}_1^b \\\\ \mathbf{x}_2^b \end{pmatrix} = \begin{pmatrix} 1 \\\\ 1 \\\\ 0 \\\\ 1 \end{pmatrix} \in \{0, 1\}^{nd},
> $$
> where we can **apply proposition 2 in a block-wise manner**.
>
> ---
>
> ### **Updates**
> We have emphasized the **block-wise application of Proposition 3.2** and updated the relevant equations, particularly **Equation (8)**, to clarify this statement and reduce ambiguity.
>
> $$$$
>
> **[W5]** *Regarding the binary quantization, I do not understand the rationale. ... Maybe the authors mean something different than what they actually wrote in Equation (9). Because of the quantization step in Equation (7), $\langle \mathbf{x}, \mathbf{y} \rangle$ is not proportional to $\langle \mathbf{x}\_{int}, \mathbf{y}_{int} \rangle$.*
>
> ---
>
> ### **Response to W5**
> We apologize for the confusion caused by omitting some intermediate steps and reducing the rigor in our explanation. To clarify, our intended statement is that $\langle \mathbf{x}^+, \mathbf{y}^+ \rangle$ is proportional to $\langle \mathbf{x}\_{int} \mathbf{y}_{int} \rangle$, assuming no quantization error.
>
> ---
>
> ### **Updates**
> To address this, we have added the missing intermediate steps and updated the corresponding content, including **Equations (8)** and **(9)** to improve clarity.
>
> $$$$
>
> **[W6]** *It seems that the authors are reinventing base-2 integer arithmetic. Any modern CPU/GPU can perform* $\langle \mathbf{x}\_{int}, \mathbf{y}\_{int} \rangle$ *with extreme efficiency at the hardware level using intrinsic SIMD instructions.*
>
> ---
>
> ### **Response to W6**
> We appreciate the reviewer for highlighting the role of SIMD instructions
>
> 1. **SIMD Benefits**
>    Our proposed Equation (10) indeed benefits from SIMD optimization. For example, on AVX512 SIMD hardware, a batch size of 512 can be used to fully exploit CPU cycles, ensuring no computational resources are wasted.
>
> 2. **FAISS Integration**
>    Additionally, our binary embeddings integrate seamlessly with FAISS, which is optimized for SIMD instructions. Unlike **integer-based indexes** that require additional transformations, our binary embeddings can be directly used with FAISS. As shown in **Table 2**, yielding a **~4x acceleration** for 256-bit binary hyperbolic embeddings.
>
> 3. **Integers have less flexibility**
>    While our method allows quantizing with any number of bits, integers only support 8*k bits due to hardware limit.
>
> ---

---

> > ### Author Response · Authors · 2024-11-25
> > **Response to Reviewer Fcqp (Other Questions)**
> >
> > **[Q4]** *Please evaluate whether Proposition 3.1 has a simpler alternative in line with the arguments I provided.*
> >
> > ---
> >
> > ### **Response to Q4**
> > We appreciate the suggestion of the alternative $1 - \cos(q, x)$. However, it has both theoretical and practical shortcomings compared to our proposed solution:
> >
> > 1. **Theoretical Shortcomings**
> >    The alternative lacks **ranking preservation**, meaning it cannot distinguish between queries with different lengths. This distinction is critical in the Poincaré ball model, where vector norms hold significant information.
> >
> > 2. **Practical Shortcomings**
> >    Integer-based approaches are not directly supported by FAISS in some cases, requiring intermediate transformations to incorporate FAISS efficiently.
> >
> > ---
> >
> > **[Q5]** *Please evaluate whether binarization (Proposition 3.2 and Section 3.3) is even needed, as working with integers seems to be more efficient than decomposing their arithmetic.*
> >
> > ---
> >
> > ### **Response to Q5**
> > While integers can be efficient, they are less flexible. Existing libraries support only standard 8/16/32/64-bit integers, whereas our approach often uses **2 bits per dimension** for optimal performance.
> >
> > We favor binary representations not only for their **theoretical simplicity** but also for their ability to integrate seamlessly with the FAISS library, which has been crucial for achieving the results shown in our experiments.
> >
> > ---
> >
> > **[Q6]** *Please clarify how the training is done regarding the data (separation into training/evaluation/test sets, etc.).*
> >
> > ---
> >
> > ### **Response to Q6**
> > - **Supervised Setting**
> >   The training pipeline follows Long et al. (2020). Specifically, the backbone network is kept fixed, and only the hyperbolic head is trained. Dataset splits are based on the standard training/validation splits for ImageNet and Moment-in-Time, and the training/testing splits for CIFAR100.
> >
> > - **Unsupervised Setting**
> >   The training pipeline aligns with [r1], with the hashing head removed.
> >
> > ---
> >
> > **[Q7]** *The common operation in scalar quantization is to perform rounding (e.g., floor(x + 0.5)) instead of truncation as it yields a lower quantization error. Why are the authors using truncation in Equation (7)?*
> >
> > ---
> >
> > ### **Response to Q7**
> > In Equation (7), we indeed mean **rounding**. We apologize for the confusion caused by the notation and have updated the manuscript to use the more commonly accepted rounding operator $\lfloor \cdot \rceil$ instead of $\lfloor \cdot \rfloor$.
> > ```

---

> ### Author Response · Authors · 2024-12-04
> **### Final Round Response to Reviewer Fcqp**
>
> **[Final Round W1.1]:** *"... presents unsupervised experiments ... The authors should pick strong baselines ..."*
>
> First, we have followed the reviewer's suggestions by clearly stating that our work focuses on the **supervised setting** in the **introduction, experiment, and implementation details** sections. Additionally, we have updated the **title** to emphasize our focus on supervised learning.
>
> Second, our unsupervised approach is limited to self-supervised learning over features and does not fully leverage the power of unsupervised learning. Although our end-to-end experiments show improvements over these results, as we are now focusing on the supervised setting, we will not discuss the unsupervised experiments in this version.
>
> ---
>
> **[Final Round W1.2]:** *"... Regarding memory savings ... I see no improvement over the baseline."*
>
> Since we are now focused on the supervised setting, we direct the reviewer to the FAISS results in Table 2 and the supervised setting of PQ for a clearer comparison.
>
> ---
>
> **[Final Round W2]:** *"... it is clearly possible to store the norm of each vector in memory at the cost of 2 bytes per vector ... computational cost equivalent"*
>
> We respectfully point out that this statement is incorrect. The reviewer might mean **2 × dim bytes per vector**, which is still **8×** our memory overhead for the 2-bit setting or **4×** for the 4-bit setting.
>
> For the speed concerns, we remove all the **slowness descriptions** as they don't influence our contribution to getting compacter and faster.
>
> ---
>
> **[Final Round W3]:** *"... Regarding the claims about scalability ... There are many standard datasets ..."*
>
> We appreciate the reviewer for suggesting datasets. For scalability, we refer to Table 7, where we demonstrate the ability to handle datasets of approximately **10 million samples**, comparable to the scales of the datasets provided by the reviewer (e.g., text2image-10M, yfcc-10M, msturing-10M-clustered).
>
> However, we respectfully note that the suggested datasets are tailored for standard Approximate Nearest Neighbor (ANN) settings, which simplify labels significantly. This is fundamentally different from our setting, as the suggested datasets treat the lowest distance pairs as the ground truth, rather than considering actual query-response relevance.
>
> Although our method does not perfectly maintain the exact nearest neighbor due to quantization, the deviation from the true nearest neighbor is minimal, as shown on **text2image-10M**:
>
> | Bit Precision | Standard Deviation | Mean Distance | StdDev / Mean |
> |---------------|---------------------|---------------|---------------|
> | 2 bit         | 21.81              | 841.07        | 2.5931%       |
> | 4 bit         | 31.23              | 1763.53       | 1.7709%       |
> | 8 bit         | 104.52             | 105671.12     | 0.0989%       |
>
> ---
>
> **[Final Round W4]:** *"How are the authors quantizing the data (some numbers are rounded up, and some numbers are rounded down) and yet maintaining proportionality?"*
>
> We respectfully point out that this is a theoretical result based on the **assumption of zero quantization error**, as explained in **Line 237** accompanying **Equation (8)**. In practice, the quantization incurs slight distance deviations, which does little harm for retrieval outputs, as suggested in Table 2.
>
> ---
>
> We especially thank reviewer Fcqp's for the scrutiny, which helped improve our paper a lot.  We still request to say that we are not an ANN paper, thus, ANN settings do not apply to us, we also request the reviewer to view this paper from an angle that is different from ANNs, and also consider Reviewer sdRm's comments on ``The paper is highly original and novel, with well-substantiated arguments and no overclaims.``

---

### Official Review · Reviewer_dMFx · 2024-11-04

**Soundness:** 3
**Presentation:** 2
**Contribution:** 2
**Rating:** 5
**Confidence:** 4

**Summary:**

Paper introduces a new binary quantizater scheme for hyperbolic embeddings to capture similarity.
Traditional models use Deep Learning machinery to produce "euclidean vectors" for objects so that the L2 distance between vectors is smaller for more similar documents.

Recently over last few years, there has been work on exploring non euclidean geometries. Namely, hyperbolic geometries and distances has emerged as a way to capture similarities, especially hierarchies of "semantic concepts" are seemingly better captured by hyperbolic embeddings.

This paper shows that we can effectively "quantize" each dimension using a simple linear quantizer, and then use hamming distance to rank the vectors to retrieve similarity based on original hyperbolic distances.
They compare the quality of their embedding, retrieval speed, and other metrics on 2-3 datasets to show that the current method outperforms good and competitive baselines.

**Strengths:**

* The hyperbolic geometrics are increasingly becoming more popular, and this paper is timely in that regard
* paper has interesting theory and empirical results
* experiments are well thought out and cover many ablation studies to prove their case

**Weaknesses:**

Proposition 3.1  and 3.2 talk about preserving rankings. Not about how close the distances are after the transformations.
In particular, it could be that original distances are 10 and 20 and 25, and after transformation, become 10, 11, and 12. This is particularly important as for large-scale datasets, we often use approximate nearest neighbor search and not exact search. So ranking is less important than the preserving the distance gaps.

Abstract says your approach is modular and can be combined with other methods like PQ, but not shown any details. There are comparisons with PQ but not detailed how to combine the two methods.

The datasets compared for benchmarks are somewhat dated, and not using the latest models/ embeddings. Maybe these are standard in hyperbolic literature, but not clear how they predict the applicability in newer LLM-based models like LLama, GritLM, etc.

There is quality comparison with PQ, but not speed comparisons.

There could be a more gentler introduction into hyperbolic geometries, and why they are useful. It seems to suddenly be thrown at the reader. Of course, this matters only for the unfamiliar reader. Also how the hierarchy of concepts topic is covered is a little abrupt and out of place.

**Questions:**

Line 62: "under retrieval" what this means?

Line 144: Why is the computational complexity high? Elaborate more

Line 148: Is this symmetric distance? Is there a rewrite which makes it symmetric?

Line 197: "five models": elaborate which five, many readers might be unaware.

Table 2: Retrieval speed: is it per query, or total across N queries for some N. Does it include the neural network time, or just the ranking of binary embeddings

Table 1: What is S^d?

4.4: Line 417: The paragraph is not clear. Make it more elaborate.

Can we combine with LSH type quantization methods instead of simple linear scaling of the values into bits?

---

> ### Author Response · Authors · 2024-11-24
> **Response to Reviewer dMFx(1/3)**
>
> ## Reponse to Reviewer dMFx (1/3)
> $$$$
>
> We sincerely thank you for recognizing the **timeliness** of our work on hyperbolic geometry and for your positive assessment of the **theoretical contributions and empirical results**. We are delighted that you found our experiments and ablation studies to be **well-designed and comprehensive**.
>
> $$$$
>
> We address your concern as follows:
> $$$$
>
> [W1] **Propositions 3.1 and 3.2 focus on preserving rankings, but ranking is less important than preserving distance gaps. This is particularly relevant for large-scale datasets, where approximate nearest neighbor (ANN) search is often used instead of exact search.**
>
> **Response to [W1]:** Thank you for raising this important point, as **it turns out to be our strength**. Our approach has capability to perform **exact nearest neighbour search**, which eliminates the reliance on approximate nearest neighbor (ANN) methods. By leveraging our compact low-bit representations, one can achieve exhaustive comparisons of 1 billion vectors in just 64GB of memory on a single computation node.
>
> In this context, **distance gap preservation becomes less critical**, as exact search inherently ensures precise comparisons across all relevant vectors. This direct comparison allows us to focus on preserving rankings while maintaining computational efficiency, which avoids the compromises typically associated with ANN methods.
>
> - We do not prove distance gap equivalence also because to achieve gap equivalence, our method would degenerate to the case $c=0$, which is euclidean, such that we lose all the benefit of being hyperbolic.
> - Note that in Propositions 3.1 and 3.2, ranking preservation applies not only between the query and the response but also between data point pairs. Therefore, nearby points stay nearby and far-away points stay far away.
>
> **[Update]: We have updated the discussion in the related work section on **Approximate Nearest Neighbor** to address this distinction and highlight the advantages of our approach.**
> $$$$
>
>
> [W2] **There are comparisons with PQ, but it is not detailed how the two methods are combined.**
>
> **Response to [W2]:** We simply use *binary hyperbolic embeddings* as input feature vectors to PQ, which not only improves performance compared to using *real-valued hyperbolic embeddings*.
> This improvement might be due to the binary grouping in Eq. (8), which already organizes the data into subspaces that align well with PQ's subspace division.
>
> **[Update]: We have included this discussion in Table 5 and Section 4.6.**
> $$$$
>
> [W3] **The datasets compared for benchmarks are somewhat dated, Maybe these are standard in hyperbolic literature, but not clear how they predict the applicability in newer LLM-based models like LLama, GritLM, etc.**
>
> **Response to [W3]:** Thank you for your thoughtful comment. We acknowledge that the datasets used in our benchmarks, while standard in hyperbolic literature, may not fully demonstrate applicability to newer LLM-based models such as LLaMA or GritLM. However, to the best of our knowledge, there have not been hyperbolic LLMs prior to our work that employ binarization. The most related works, such as Hyperbolic Fine-tuning for Large Language Models [r1], focus on hyperbolic parameter spaces (e.g., for low-rank adapters) rather than embedding spaces, which remain in Euclidean space. Similarly, Hyperbolic Pre-Trained Language Models (HPTLM) [r2] employ hyperbolic geometry but rely on an unreleased codebase, making direct comparisons with their framework currently infeasible. We intend to revisit this avenue when their implementation becomes publicly available.
>
> [r1] Yang, Menglin, et al. "Hyperbolic Fine-tuning for Large Language Models." arXiv preprint arXiv:2410.04010 (2024).
>
> [r2] Chen, Weize, et al. "Hyperbolic Pre-Trained Language Model." IEEE/ACM Transactions on Audio, Speech, and Language Processing (2024).
>
> **[Update]: We include this discussion in the Experimental Setup section.**
>
> We would greatly appreciate it if you could suggest a few newer and interesting datasets that you believe would better align with modern benchmarks. Your insights would help us refine our benchmarks and ensure broader applicability to these emerging models.
>
> $$$$
>
> [W4] **A more gentler introduction into hyperbolic geometries, and why they are useful, for the unfamiliar reader.**
>
> **Response to [W4]:** We briefly introduced hyperbolic geometry at the beginning of Section 3 to guide readers to the appendix.
>
> **[Update]: We updated a preliminary section in Appendix A. where we can find the introduction to hyperbolic operations and distance, symmetry, and isometry.**

---

> ### Author Response · Authors · 2024-11-24
> **Response to Reviewer dMFx(2/3)**
>
> ## Reponse to Reviewer dMFx (2/3)
>
> $$$$
>
> We answer your questions as follows:
>
> [Q1] **Line 62: "under retrieval" what this means?**
>
> [A1]: Under retrieval means we focus on ranking equivalence (ranking preservation) rather than distance equivalence.
>
> $$$$
>
> [Q2] **Line 144: Why is the computational complexity high? Elaborate more**
>
> [A2]: We have quantified the slowness in Appendix D, to have a more theoritical comparison on computational complexities of $\mathrm{d_\mathbb{D}}(\mathbf{q}, \mathbf{x})$ and $\mathrm{d_\mathbb{R}}(\mathbf{x}, \mathbf{y})$, we analyze the number of operations required for each, including constant factors.
>
> 1. **Euclidean Distance $\mathrm{d_\mathbb{R}}(\mathbf{x}, \mathbf{y})$:**
>    $$
>    \mathrm{d_\mathbb{R}}(\mathbf{x}, \mathbf{y}) = \|| \mathbf{x} - \mathbf{y} \||^2
>    $$
>    - **Subtractions:** $n$ (computing $\mathbf{x} - \mathbf{y}$)
>    - **Squaring components:** $n$ multiplications
>    - **Summing squares:** $n - 1$ additions
>    - **Total operations:** $2n - 1$ additions/subtractions, $n$ multiplications
>
> 2. **Distance $\mathrm{d_\mathbb{D}}(\mathbf{q}, \mathbf{x})$:**
>    $$
>    \mathrm{d_\mathbb{D}}(\mathbf{q}, \mathbf{x}) = \frac{2}{\sqrt{c}} \tanh^{-1} \left( \sqrt{c} \left\|\left| \frac{(1 - 2c \langle \mathbf{q}, \mathbf{x} \rangle + c \|\|\mathbf{x}\|\|^2) \mathbf{q} + (1 - c \||\mathbf{q}\||^2) \mathbf{x}}{1 - 2c \langle \mathbf{q}, \mathbf{x} \rangle + c^2 \|\mathbf{q}\||^2 \||\mathbf{x}\||^2} \right\|\right\|^2 \right)
>    $$
>    - **Inner products and norms:**
>      - $\langle \mathbf{q}, \mathbf{x} \rangle$: $n$ multiplications, $n - 1$ additions
>      - $\|\mathbf{q}\|^2$ and $\|\mathbf{x}\|^2$: $2n$ multiplications, $2(n - 1)$ additions
>    - **Scalar computations:** Few multiplications and additions (constants)
>    - **Vector operations:**
>      - Scaling vectors: $2n$ multiplications
>      - Vector addition: $n$ additions
>    - **Norm squared of numerator:** $n$ multiplications, $n - 1$ additions
>    - **Division:** One division (scalar)
>    - **Transcendental function:** One $\tanh^{-1}$ evaluation
>    - **Total Operations:**
>      - **Multiplications:** Approximately $7n + \text{constants}$.
>      - **Additions/Subtractions:** Approximately $5n + \text{constants}$.
>      - **Divisions:** and **Transcendental Functions:** $1 \text{ each}$.
>
>
> **Conclusion:**
>
> - While both distances have linear complexity $O(n)$, $\mathrm{d_\mathbb{D}}(\mathbf{q}, \mathbf{x})$ involves significantly more arithmetic operations.Therefore, even when considering constant factors, $\mathrm{d_\mathbb{D}}(\mathbf{q}, \mathbf{x})$ is computationally more intensive than $\mathrm{d_\mathbb{R}}(\mathbf{x}, \mathbf{y})$.
>
> $$ $$
>
> [Q3] **Line 148: Is this symmetric distance? Is there a rewrite which makes it symmetric?**
>
> [A3]: **[Update]: We give a detailed induction on how the distance satisfies symmety in Appendix A.5.**
>
> Besides, we can also view this in a lorentzian point-of-view.
> While poincare distance with curvature $c$ seemed complex, we can see the symmetry in its isometry, distance under Lorentzian model, defined as:
>
> $$
> d_{\mathcal{L}}^c(\mathbf{x}, \mathbf{y})=\frac{1}{\sqrt{|c|}} \cosh ^{-1}\left(c\langle\mathbf{x}, \mathbf{y}\rangle_{\mathcal{L}}\right),
> $$ where $\langle\cdot, \cdot\rangle_\mathcal{L}$ is the Lorentzian inner product:
>
> $$
> \langle \mathbf{x}, \mathbf{y}\rangle_\mathcal{L}=-x_0 y_0+x_1 y_1+x_2 y_2+\cdots+x_n y_n
> $$
>
> We can see that $\langle \mathbf{x}, \mathbf{y}\rangle_\mathcal{L} = \langle \mathbf{y}, \mathbf{x}\rangle_\mathcal{L}$, thus $d_{\mathcal{L}}^c(\mathbf{x}, \mathbf{y})=d_{\mathcal{L}}^c(\mathbf{y}, \mathbf{x})$.

---

> ### Author Response · Authors · 2024-11-24
> **Response to Reviewer dMFx(3/3)**
>
> ## Response to Reviewer dMFx(3/3)
>
> [Q4] **Line 197: "five models": elaborate which five, many readers might be unaware.**
>
> [A4]: **[Update]: We give detailed clarification in Appendix A.2 on the isometry.**
>
>
> We follow the convention in differential geometry, that five commonly used hyperbolic models are isometry to each other:
> - H: Halfspace model (Poincare Halfspace).
> $$ H= \{ (1, x_2, \ldots, x_{n+1} ): x_{n+1}>0 \} $$
> - I: Interior of the disk model (Poincare disk).
> $$ I= \{\left(x_1, \ldots, x_n, 0\right): x_1^2+\cdots+x_n^2<1 \} $$
> - J: Jemisphere model (pronounce the J as in Spanish).
> $$  \{\left(x_1, \ldots, x_{n+1}\right): x_1^2+\cdots+x_{n+1}^2=1 \text { and } x_{n+1}>0 \} $$
> - K: Klein model.
> $$
> K= \{\left(x_1, \ldots, x_n, 1\right): x_1^2+\cdots+x_n^2<1 \}
> $$
> - L: Loid model (short for hyperboloid).
> $$
> L= \{\left(x_1, \ldots, x_n, x_{n+1}\right): x_1^2+\cdots+x_n^2-x_{n+1}^2=-1 \text { and } x_{n+1}>0 \}
> $$
> Isometry implies that their distance metric are equivalent to each other under coordinate transformations defined in section 7 of [r3].
>
> [r3] Cannon, James W., et al. "Hyperbolic geometry." Flavors of geometry 31.59-115 (1997): 2.
>
> [Q5] **Table 2: Retrieval speed: is it per query, or total across N queries for some N. Does it include the neural network time, or just the ranking of binary embeddings.**
>
> [A5]: It's the total across N queries, where N is the size of the query set. It does not include neural network time, as neural networks only embed the query at a negligible time compared with the calculating distance and ranking over the entire database.
>
> $$$$
>
> [Q6] **Table 1: What is S^d?**
>
> [A6]: $\mathbb{S}^d$ defines the hyperspherical space with $d$ dimentionality such that $\mathbb{S}^d = \{\mathbf{x}, \|| \mathbf{x}\|| = r\}$.
>
> **[Update]:To calrify, we update the caption in Table 1 accordingly.**
>
> $$$$
>
> [Q7] **Line 417: The paragraph is not clear. Make it more elaborate.**
>
> [A7]: We update it as this:
>
> A key benefit of hyperbolic space is the capability to embed hierarchical knowledge with minimal distortion for hierarchical embeddings Ganea et al. (2018b);
> Sala et al. (2018). Such property enables hyperbolic network to retrieve semantically similar items of adjacent classes Long et al.(2018). Following this setting, in Appendix E.2 we report results with hierarchical
> knowledge-aware retrieval. we are able to maintain retrieval performance both in standard setting and hierarchy-aware setting. Even with a lowest number of bit length.
>
> $$$$
>
> [Q8] **Can we combine with LSH type quantization methods instead of simple linear scaling of the values into bits?**
>
> [A8]: LSH-based quantization is one potential improvement over our method; we leave it for future research directions.

---

### Meta-Review · Area_Chair_7gdG · 2024-12-19

**Metareview:**

Thanks for your submission to ICLR.

This paper received mixed reviews (two negative and two positive).  The reviewers had various concerns, including multiple issues with the theory, questions about the computational cost, questions about the experimental results, and issues with the writing/presentation.

After the rebuttal, there was some discussion amongst reviewers and AC on this paper.  The most negative reviewer still maintained many of their concerns.  In particular, they continued to disagree about several points, including:

-Regarding computational cost, the reviewer pointed out that storing the norm is independent of the dimensionality, counter to the response during the rebuttal.  I agree that this was not adequately addressed by the authors.

-The reviewer maintained concerns about the theory, including issues with quantization in Prop 3.2.

-The reviewer maintains that the introduction of the binary embedding is not contributing meaningfully to the paper.

-The reviewer is concerned that being restricted to the supervised setting limits the applicability and contribution of the paper.

Unfortunately, the other negative reviewer did not participate in the discussion of the paper.  However, overall in looking at the criticisms of the paper, I tend to agree with the more negative reviewers that, at a minimum, this paper needs more work before it is ready for publication.  There are too many unresolved issues still, with not enough enthusiasm for accepting the paper amongst the reviewers.

**Additional Comments On Reviewer Discussion:**

See the discussion above.  In the discussion, the most negative reviewer continued to maintain that there were several issues with the paper, and none of the positive reviewers were willing to champion the paper.

---

### Decision · Program_Chairs · 2025-01-22

Reject